# PAC-Bayes Generalization Certificates for Learned Inductive Conformal Prediction

**Apoorva Sharma**
NVIDIA Research
apoorvas@nvidia.com

**Sushant Veer**
NVIDIA Research
sveer@nvidia.com

**Asher Hancock**
Princeton University
ah4775@princeton.edu

**Heng Yang**
Harvard University
& NVIDIA Research
hengy@nvidia.com

**Marco Pavone**
Stanford University
& NVIDIA Research
mpavone@nvidia.com

**Anirudha Majumdar**
Princeton University
ani.majumdar@princeton.edu

## Abstract

Inductive Conformal Prediction (ICP) provides a practical and effective approach for equipping deep learning models with uncertainty estimates in the form of set-valued predictions which are guaranteed to contain the ground truth with high probability. Despite the appeal of this coverage guarantee, these sets may not be efficient: the size and contents of the prediction sets are not directly controlled, and instead depend on the underlying model and choice of score function. To remedy this, recent work has proposed learning model and score function parameters using data to directly optimize the efficiency of the ICP prediction sets. While appealing, the generalization theory for such an approach is lacking: direct optimization of empirical efficiency may yield prediction sets that are either no longer efficient on test data, or no longer obtain the required coverage on test data. In this work, we use PAC-Bayes theory to obtain generalization bounds on both the coverage and the efficiency of set-valued predictors which can be directly optimized to maximize efficiency while satisfying a desired test coverage. In contrast to prior work, our framework allows us to utilize the entire calibration dataset to learn the parameters of the model and score function, instead of requiring a separate hold-out set for obtaining test-time coverage guarantees. We leverage these theoretical results to provide a practical algorithm for using calibration data to simultaneously fine-tune the parameters of a model and score function while guaranteeing test-time coverage and efficiency of the resulting prediction sets. We evaluate the approach on regression and classification tasks, and outperform baselines calibrated using a Hoeffding bound-based PAC guarantee on ICP, especially in the low-data regime.

## 1 Introduction

Machine learning (ML) models have rapidly proliferated across numerous applications, including safety-critical ones such as autonomous vehicles [Schwarting et al., 2018, Waymo, 2021], medical diagnosis [Ahsan et al., 2022], and drug discovery [Vamathevan et al., 2019]. Due to the severity of outcomes in these applications, decision making cannot solely hinge on "point" predictions from the ML model, but must also encapsulate a measure of the uncertainty in the predictions. As a result, accurate uncertainty estimation is a cornerstone of robust and trustworthy ML systems; however, overly-optimistic or overly-conservative estimates limit their usefulness. In this paper, we will present a method that builds upon inductive conformal prediction (ICP) [Vovk et al., 2005] and probably approximately correct (PAC) Bayes theory [McAllester, 1998] to furnish uncertainty estimates that *provably* control the uncertainty estimate's conservatism while achieving desired correctness rates.

37th Conference on Neural Information Processing Systems (NeurIPS 2023).

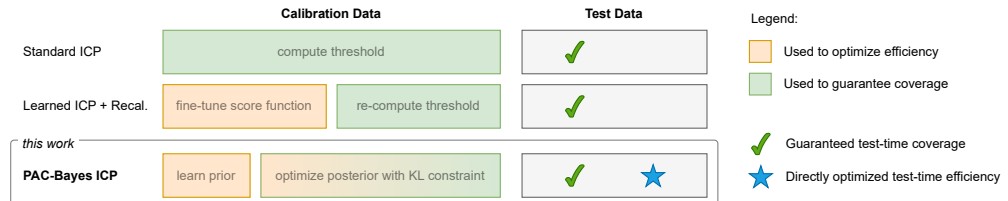

Figure 1: Standard ICP uses a calibration dataset to guarantee coverage on exchangeable test data, but the efficiency of the resulting prediction sets is not directly optimized. Using the same data to select score function and model parameters as well as to perform ICP no longer provides guarantees on test-time coverage and efficiency. Therefore, prior work on optimal ICP has typically relied on holding out some calibration data for a post-optimization recalibration step to retain coverage guarantees. In this work, we leverage PAC-Bayes theory to optimize efficiency and provide generalization guarantees using the same dataset. Practically, we demonstrate that a hybrid approach using some data to tune a data-dependent prior, and use the rest to further optimize and obtain generalization guarantees yields the best performance.

ICP has emerged as a practical approach for equipping deep learned models with uncertainty estimates in the form of set-valued predictions with high-probability coverage guarantees, i.e., with high probability, at test time, the ground truth labels will lie within the predicted set. However, post-hoc application of ICP can often lead to overly-conservative set sizes; the tightness of the predicted set (referred to as efficiency) highly depends on the underlying model as well as the choice of the score function used in ICP. Various approaches have worked towards alleviating this challenge either by developing new application-specific score functions [Romano et al., 2020, Yang and Pavone, 2023, Lindemann et al., 2023] or by optimizing the model and / or score function directly [Yang and Kuchibhotla, 2021, Cleaveland et al., 2023, Bai et al., 2022]. The direct optimization approach is appealing as a general application-agnostic method for obtaining tight uncertainty estimates; however, it requires re-calibration on held-out data in order to retain generalization guarantees on coverage. By coalescing ICP with PAC-Bayes, we are able to train using all available data, while retaining generalization guarantees for coverage *and* efficiency; see Fig. 1 for more details.

**Contributions:**   In this work, we make the following core contributions:

1. We use PAC-Bayes theory to obtain generalization bounds on both the coverage and efficiency of set-valued predictors. In contrast to prior work (see Fig. 1 and Sec. 3), our framework allows us to utilize the *entire* calibration dataset to learn the model parameters and score function (instead of requiring a held-out set for obtaining a guarantee on coverage).

2. We leverage these theoretical results to provide a practical algorithm (see Alg. 1) for utilizing calibration data to fine-tune the parameters of the model and score function while guaranteeing test-time coverage and efficiency. This algorithm allows the user to specify a desired coverage level, which in turn specifies the degree to which efficiency can be optimized (since there is a trade-off between the desired coverage level and efficiency).

3. We evaluate our approach on regression and image classification problems with neural network-based score functions. We show that our method can yield more efficient predictors than prior work on learned conformal prediction calibrated using a Hoeffding-bound based PAC guarantee on coverage [Vovk, 2012, Prop 2a].

## 2   Background: Inductive Conformal Prediction

We consider a supervised learning setup, wherein our goal is to predict labels $y \in \mathcal{Y}$ given inputs $x \in \mathcal{X}$. In inductive conformal prediction (ICP) [Vovk et al., 2005], our goal is to develop a set-valued predictor which maps inputs $x$ to a subset of the label space $C(x) \subseteq \mathcal{Y}$. Specifically, we want this set-valued function to satisfy a *coverage* guarantee, ensuring that the chance that the prediction set fails to contain the ground truth label is bounded to a user-specified level $\alpha$, i.e.

$$\mathbb{P}_{x,y \sim \mathcal{D}} (y \notin C(x)) \le \alpha, \tag{1}$$

where $\mathcal{D}$ is the joint distribution over inputs and labels that test data are drawn from. In practice, $\mathcal{D}$ is not known, but we assume we have access to a calibration dataset $D_{\text{cal}} = \{(\boldsymbol{x}_i, \boldsymbol{y}_i)\}_{i=1}^{N}$ where each example $(\boldsymbol{x}_i, \boldsymbol{y}_i)$ is exchangeable with the test data. Given this set-up, ICP constructs a set-valued predictor of the form

$$C(\boldsymbol{x}; \tau) := \{\boldsymbol{y} \in \mathcal{Y} \mid s(\boldsymbol{x}, \boldsymbol{y}) \leq \tau\}, \tag{2}$$

i.e., the $\tau$ sub-level set of the *nonconformity function* $s : \mathcal{X} \times \mathcal{Y} \to \mathbb{R}$ evaluated at the input $\boldsymbol{x}$. The nonconformity function assigns a scalar value quantifying how poorly an example $\boldsymbol{x}, \boldsymbol{y}$ conforms to the training dataset. It is common to define a nonconformity function that depends on the learned model $f$, choosing $s(\boldsymbol{x}, \boldsymbol{y}) = \ell(f(\boldsymbol{x}), \boldsymbol{y})$, measuring how poorly the prediction $f(\boldsymbol{x})$ aligns with the label $\boldsymbol{y}$ [Angelopoulos and Bates, 2021]. Armed with the calibration dataset as well as the nonconformity function, all that remains to construct the set-valued predictor is to choose the proper threshold $\tau$. ICP defines a calibration strategy to choose $\tau$ that guarantees a desired miscoverage rate $\alpha$. Specifically, let $\mathcal{S}_{\text{cal}} = \{s(\boldsymbol{x}, \boldsymbol{y}) \mid \boldsymbol{x}, \boldsymbol{y} \in D_{\text{cal}}\}$ and choose $\tau^*(D_{\text{cal}}, \alpha)$ to be the $q = \lceil (n+1)(1-\alpha) \rceil / n$ quantile of the set $\mathcal{S}$. Then, so long as the score function $s$ is independent from $D_{\text{cal}}$, we have the following guarantee on the prediction sets defined by (2):

$$\alpha - \frac{1}{N+1} < \mathop{\mathbb{P}}_{D_{\text{cal}} \sim \mathcal{D}^N, (\boldsymbol{x}, \boldsymbol{y}) \sim \mathcal{D}} (\boldsymbol{y} \notin C(\boldsymbol{x}; \tau^*(D_{\text{cal}}, \alpha))) \leq \alpha. \tag{3}$$

Importantly, this probabilistic guarantee is marginal over the sampling of the test data point as well as the calibration dataset. In practice, we are given a fixed calibration dataset, and would like to bound the miscoverage rate conditioned on observing $D_{\text{cal}}$. As shown in Vovk [2012, Prop 2a], we can obtain a probably-approximately-correct (PAC) style guarantee on the ICP predictor of the form

$$\mathop{\mathbb{P}}_{D_{\text{cal}} \sim \mathcal{D}} \left( \mathop{\mathbb{P}}_{\boldsymbol{x}, \boldsymbol{y} \sim \mathcal{D}} (\boldsymbol{y} \notin C(\boldsymbol{x}; \tau^*(D_{\text{cal}}, \alpha)) < \alpha + \sqrt{\frac{-\log \delta}{2N}} \right) \geq 1 - \delta. \tag{4}$$

In other words, with probability at least $1 - \delta$ over the sampling of a fixed calibration dataset, we can ensure the test-time miscoverage rate is bounded by $\alpha$ by calibrating for a marginal coverage rate of $\hat{\alpha} = \alpha - \sqrt{-\log \delta / 2N}$. Alternatively, one can achieve the same PAC guarantee using the tighter bound in [Vovk, 2012, Prop 2b] by choosing the largest $\hat{\alpha} \in (0, \alpha)$ for which

$$\delta \geq I_{1-\alpha}(N - \lfloor \hat{\alpha}(N+1) - 1 \rfloor, \lfloor \hat{\alpha}(N+1) - 1 \rfloor + 1) \tag{5}$$

where $I_x(a, b)$ is the regularized incomplete beta distribution. This lacks an analytic solution for the optimal $\hat{\alpha}$, but is less conservative.

The strength of ICP lies in its ability to guarantee test-time coverage for *any* base predictor and nonconformity score function. However, ICP does not directly control the efficiency of the resulting prediction sets; the size and make-up of the prediction set for any given input $\boldsymbol{x}$ is highly dependent on the score function $s$ (and thus also the base prediction model $f$). For this reason, naïve application of ICP can often yield prediction sets that are too "loose" to be useful downstream. Often, both the model and the score function may depend on various parameters $\theta \in \mathbb{R}^d$, e.g., the weights of a neural network base prediction model. The values of these parameters influence the ICP procedure, from the computation of the score function $s(\boldsymbol{x}, \boldsymbol{y}; \theta)$, the resulting threshold $\tau^*(D_{\text{cal}}, \alpha; \theta)$, and finally, the prediction sets themselves $C(\boldsymbol{x}; \tau^*, \theta)$. Our goal in this work is to investigate how we can use calibration data to fine-tune these parameters to optimize an efficiency objective while retaining guarantees on test-time coverage.

## 3   Related Work

**Handcrafted score functions.** In classification ($\mathcal{Y} = [K]$), suppose $f(\boldsymbol{x})$ predicts the softmax class probabilities; typically the score function is chosen as either $s(\boldsymbol{x}, y) = 1 - f(\boldsymbol{x})_y$ or $s(\boldsymbol{x}, y) = \sum_{k \in [K]} \{f(\boldsymbol{x})_k \mid f(\boldsymbol{x})_k \geq f(\boldsymbol{x})_y\}$, where $f(\boldsymbol{x})_y$ denotes the softmax probability of the groundtruth label. While the former score function produces prediction sets with the smallest average size [Sadinle et al., 2019], the latter produces prediction sets whose size can adapt to the difficulty of the problem [Romano et al., 2020]. In 1-D regression ($\mathcal{Y} = \mathbb{R}$), [Romano et al., 2019] recommend training $f(\boldsymbol{x}) = [f_{\alpha/2}(\boldsymbol{x}), f_{1-\alpha}(\boldsymbol{x})]$ using quantile regression to output the (heuristic) $\alpha/2$ and $1 - \alpha/2$ quantiles, and then set the score function as $s(\boldsymbol{x}, y) = \max\{f_{\alpha/2}(\boldsymbol{x}) - y, y - f_{1-\alpha}(\boldsymbol{x})\}$

to compute the distance from the groundtruth $y$ to the prediction interval $f(\boldsymbol{x})$. In $n$-D regression ($\mathcal{Y} = \mathbb{R}^n$), it is common to design $s(f(\boldsymbol{x}), \boldsymbol{y}) = \frac{\|\mu(\boldsymbol{x}) - \boldsymbol{y}\|}{u(\boldsymbol{x})}$, where $f(\boldsymbol{x}) = (\mu(\boldsymbol{x}), u(\boldsymbol{x}))$ outputs both the mean of the label $\mu(\boldsymbol{x})$ and a heuristic notion of uncertainty $u(\boldsymbol{x})$ [Yang and Pavone, 2023, Lindemann et al., 2023]. While this leads to a ball-shaped prediction set, designing $s(f(\boldsymbol{x}), \boldsymbol{y}) = \max_{i \in [n]} \{|\mu(\boldsymbol{x})_i - y_i|\}$ can produce a box-shaped prediction set [Bai et al., 2022].

**Learning score functions.** [Yang and Kuchibhotla, 2021] propose selection algorithms to yield the smallest conformal prediction intervals given a family of learning algorithms. [Cleaveland et al., 2023] consider a time series prediction setup and leverage linear complementarity programming to optimize a score function parameterized over multiple time steps. [Stutz et al., 2021] develop conformal training to shrink the prediction set for classification, which learns the prediction function and the score function end-to-end by differentiating through and simulating the conformal prediction procedure during training. [Bai et al., 2022] considers optimizing the efficiency of a parameterized score function subject to coverage constraints on a calibration dataset, and derives generalization bounds depending on the complexity of the parametrization. [Einbinder et al., 2022] focus the classification-specific adaptive predictive set (APS) score function [Romano et al., 2020] score function, and propose an auxiliary loss term to train the model such that its predictions more closely meet the conditions under which the APS score function yields optimal efficiency. The drawback of [Cleaveland et al., 2023, Stutz et al., 2021, Bai et al., 2022, Einbinder et al., 2022] is that a separate held-out dataset is required to *recalibrate* the optimized score function to obtain test-time coverage guarantees. In this paper, we leverage PAC-Bayes theory to alleviate this drawback and allow learning the parameters of the model and score function using the entire calibration dataset, while offering efficiency and coverage guarantees.

**Generalization theory and PAC-Bayes.** Generalization theory seeks to quantify how well a given model will generalize to examples beyond the training set, and how to learn models that will generalize well. Early work includes VC theory [Vapnik and Chervonenkis, 1968], Rademacher complexity [Shalev-Shwartz and Ben-David, 2014], and the minimum description length principle [Rissanen, 1989, Blumer et al., 1987]. In this work, we make use of PAC-Bayes theory [McAllester, 1998]. In PAC-Bayes learning, one fixes a data-independent prior over models and then obtains a bound on the expected loss that holds for any (potentially data-dependent) choice of posterior distribution over models. One can then optimize the PAC-Bayes bound via the choice of posterior in order to obtain a certificate on generalization. In contrast to bounds based on VC theory and Rademacher complexity, PAC-Bayes provides numerically strong generalization bounds for deep neural networks for supervised learning [Dziugaite and Roy, 2017, Neyshabur et al., 2017a,b, Bartlett et al., 2017, Arora et al., 2018, Rivasplata et al., 2019, Pérez-Ortiz et al., 2021, Jiang et al., 2020, Lotfi et al., 2022] and reinforcement learning [Fard et al., 2012, Majumdar et al., 2021, Veer and Majumdar, 2020, Ren et al., 2021]. We highlight that the standard framework of generalization theory (including PAC-Bayes) provides bounds for *point* predictors (i.e., models that output a single prediction for a given input). Here, we utilize PAC-Bayes in the context of conformal prediction to learn *set-valued* predictors with guarantees on coverage and efficiency.

## 4 PAC-Bayes Generalization Bounds for Inductive Conformal Prediction

Optimizing efficiency with ICP requires splitting the dataset to train on one part and calibrate on the other. The reduced data for calibration can result in weaker miscoverage guarantees. Drawing from PAC-Bayes generalization theory, we will develop a theory that facilitates simultaneous calibration and efficiency optimization for ICP using the *entire* calibration dataset.

Our generalization bounds are derived by randomizing the parameters of the score function $\theta$. Let $Q(\theta)$ be a distribution over parameters $\theta$. Note that, for a fixed target miscoverage rate $\hat{\alpha}$, every sample from $\theta$ induces a different prediction set $C(\boldsymbol{x}; \tau^*(D_{\text{cal}}, \hat{\alpha}; \theta), \theta)$. Test-time coverage for such a randomized prediction set corresponds to the probability of miscoverage, marginalizing over the sampling of $\theta$:

$$\mathcal{L}_{\text{cov}}(Q) := \mathbb{P}_{\theta \sim Q} \left( \mathbb{P}_{\boldsymbol{x}, \boldsymbol{y} \sim \mathcal{D}} (\boldsymbol{y} \notin C(\boldsymbol{x}; \tau^*(D_{\text{cal}}, \hat{\alpha}; \theta), \theta)) \right). \tag{6}$$

The *efficiency* $\ell_{\text{eff}}$ of a predicted set $C$ is a metric that can encode task-specific preferences (e.g., the volume of the predicted set) that we wish to minimize while satisfying the desired miscoverage rate.

Similar to (6), we can define the efficiency measure $\mathcal{L}_{\text{eff}}$ on randomized prediction sets by taking an expectation of a single-set efficiency measure:

$$\mathcal{L}_{\text{eff}}(Q) := \mathop{\mathbb{E}}_{\theta \sim Q} \left[ \mathop{\mathbb{E}}_{\boldsymbol{x}, \boldsymbol{y} \sim \mathcal{D}} \left[ \ell_{\text{eff}}(C(\boldsymbol{x}; \tau^*(D_{\text{cal}}, \hat{\alpha}; \theta), \theta), \boldsymbol{y}) \right] \right]. \tag{7}$$

Our core theoretical contributions are to provide bounds on $\mathcal{L}_{\text{cov}}(Q)$ and $\mathcal{L}_{\text{eff}}(Q)$ that can be computed using $D_{\text{cal}}$ and, critically, hold uniformly for *all* choices of $Q$, even those that depend on $D_{\text{cal}}$. This allows us to *learn* the distribution $Q$ over score function parameters using $D_{\text{cal}}$. First, we state our generalization bound for test-time coverage.

**Theorem 1** (PAC-Bayes Bound on Coverage of ICP Prediction Sets). *Let $P(\theta)$ be a (data-independent) prior distribution over the parameters of the score function. Let $D_N$ be a set of $N$ samples from $\mathcal{D}$. Choose empirical coverage parameter $\hat{\alpha} \in (0, 1)$ such that $N > 1/\hat{\alpha} - 1$. Then, with probability greater than $1 - \delta$ over the sampling of $D_N$, the following holds simultaneously for all distributions $Q(\theta)$:*

$$\text{kl}\left( \frac{\lfloor (N+1)\hat{\alpha} \rfloor - 1}{N - 1} \,\middle\|\, \mathcal{L}_{\text{cov}}(Q) \right) \leq \frac{\text{KL}(Q \| P) + \log\left( \frac{B(N)}{\delta} \right)}{N - 1}, \tag{8}$$

*where $\text{kl}(p \| q)$ is the KL divergence between Bernoulli distributions with success probability $p$ and $q$ respectively, and $B(N) := \text{Beta}(\frac{k-1}{N-1}, k, N+1-k = O(\sqrt{N/(\hat{\alpha}(1-\hat{\alpha})))})$, where $k = \lfloor (N+1)\hat{\alpha} \rfloor$*

The proof for this theorem is presented in App. A[1]. This theorem states that applying the calibration procedure will yield a test-time coverage rate $\mathcal{L}_{\text{cov}}(Q)$ close to $\frac{\lfloor (N+1)\hat{\alpha} \rfloor - 1}{N-1}$, and the amount that it can differ shrinks as the cardinality $N$ of the calibration set increases, but grows as $Q$ deviates from $P$. Finally, we note that an upper bound on $\mathcal{L}_{\text{cov}}(Q)$ can computed by inverting the KL bound [Dziugaite and Roy, 2017] using convex optimization [Majumdar et al., 2021, Sec. 3.1.1].

Next, we will provide a generalization bound on efficiency.

**Theorem 2** (PAC-Bayes Bound on Efficiency of ICP Prediction Sets). *Let $P(\theta)$ be a (data-independent) prior distribution over the parameters of the score function. Let $D_N$ be a set of $N$ samples from $\mathcal{D}$ and $\gamma > 0$. Let the efficiency $\ell_{\text{eff}}$ of a predicted set $C$ always lie within[2] $[0, 1]$. Assume the score function is bounded, $s(\boldsymbol{x}, \boldsymbol{y}; \theta) < \beta$, and the efficiency loss $\tau \to \ell_{\text{eff}}(C(\boldsymbol{x}; \tau, \theta)$ is $L_\tau$-Lipschitz continuous in $\tau$ for all values of $\theta$. Then, with probability greater than $1 - \gamma$, we have for all $Q$,*

$$\mathcal{L}_{\text{eff}}(Q) \leq \mathop{\mathbb{E}}_{\theta \sim Q} \left[ \frac{1}{N} \sum_{i=1}^{N} \ell_{\text{eff}}(C(\boldsymbol{x}_i; \tau^*(D_N, \hat{\alpha}; \theta), \theta)) \right] + \frac{2\beta L_\tau}{\sqrt{N}} + \frac{\sqrt{\frac{1}{2}\text{KL}(Q\|P) + \frac{1}{2}\log(\frac{2N}{\gamma})}}{\sqrt{N-1}} \tag{9}$$

The proof for this theorem follows a similar approach to standard PAC-Bayes bounds, but is complicated by the dependence of $\tau^*$ on $D_N$. To mitigate this, we instead consider the generalization error between empirical and test-time efficiency for the worst-case choice of $\tau$, which requires assuming $\tau$ is bounded and the efficiency objective smooth w.r.t. $\tau$. The full proof is presented in App. A.

## 5 Practical Algorithmic Implementation

The generalization bounds above suggest a practical algorithm for using calibration data to simultaneously conformalize a base predictor while also fine tuning model and score function parameters. Specifically, we leverage the theory to formulate a constrained optimization problem, where the objective aims to minimize an efficiency loss, while the constraint ensures that test-time coverage can still be guaranteed. The overall algorithm is summarized in Alg. 1.

---

[1]Our bound takes a similar form to the Maurer-Langford-Seeger PAC-Bayesian bound [Seeger et al., 2001, Maurer, 2004] on generalization risk given empirical risk. However, while the *risk* on two samples is independent for a fixed hypothesis, the *coverage* for a particular sample depends the threshold $\tau^*$, which is a function of the entire calibration dataset $D_N$. This key difference precludes a direct application existing bounds and necessitates a novel theoretical approach.

[2]Any uniformly bounded efficiency measure can be mapped to [0,1] by a scaling factor.

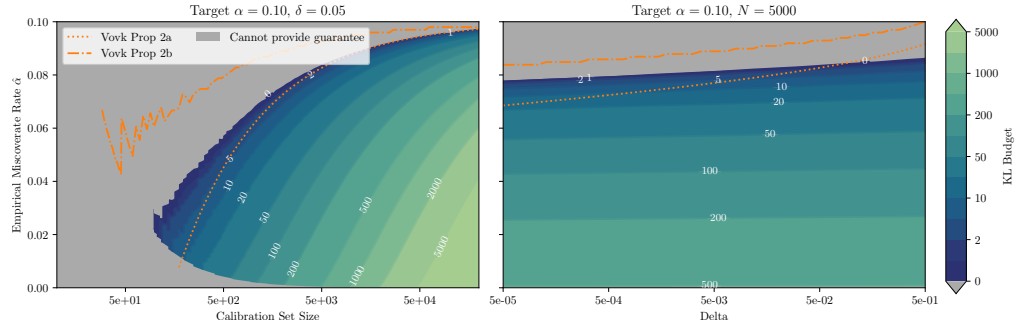

Figure 2: Visualization of the KL budget allowed by the PAC-Bayes generalization bound to achieve a target test-time coverage of $\alpha = 0.1$, plotted as a function of $\hat{\alpha}$ and calibration set size $N$ (left) as well as guarantee failure probability $\delta$ (right). Note the log scale on the x axis on both plots.

## 5.1 Guaranteeing test-time coverage via a KL constraint

Using the bound in Theorem 1, we derive a constraint over data-dependent posteriors $Q(\theta)$ such that choosing a threshold using the conformal prediction procedure $\tau(\theta, D_N, \hat{\alpha})$ will guarantee a test-time coverage greater than $1 - \alpha$, where the coverage we enforce over $D_N$, $(1-\hat{\alpha})$, maybe different from the coverage we wish to achieve at test time, $1 - \alpha$.

**Corollary 2.1** (Constraint on Data-Dependent Posterior). *Fix $\hat{\alpha} \leq \alpha$, and a prior distribution $P(\theta)$. Given a calibration dataset $D_{\text{cal}}$ of $N$ i.i.d. samples from $\mathcal{D}$, we have that with probability greater than $1 - \delta$, then simultaneously for all distributions $Q(\theta)$ which satisfy*

$$\text{KL}(Q\|P) \leq B(\alpha, \hat{\alpha}, \delta, N) := (N-1)\text{kl}\left(\frac{\lfloor (N+1)\hat{\alpha} \rfloor - 1}{N-1} \,\middle\|\, \alpha\right) - \log\left(\frac{B(N)}{\delta}\right), \quad (10)$$

*it holds that $\mathcal{L}_{\text{cov}}(Q) \leq \alpha$.*

The proof for this corollary is provided in App. A. Importantly, this bound holds for all $Q(\theta)$ that satisfy the KL constraint, including $Q$ that depend on the calibration data $D_N$. Thus, we are free to use any optimization algorithm to search for a feasible $Q$ that minimizes any auxiliary loss; with $B(\alpha, \hat{\alpha}, \delta, N)$ serving as a "budget" for optimization, limiting the degree to which $Q$ can deviate from the data-independent prior $P$. Fig. 2 visualizes this budget as a function of $N$, and $\hat{\alpha}$ for a fixed $\alpha$ and $\delta$. As the plots show, by choosing the threshold $\tau$ to attain a more conservative $\hat{\alpha}$, we can afford more freedom to vary $Q$. The curve defined by $B(\alpha, \hat{\alpha}, \delta, N) = 0$ indicates the maximum $\hat{\alpha}$ for which our theory provides a guarantee on $1 - \alpha$ test time coverage with probability greater than $1 - \delta$ for a randomized data-independent score function ($Q = P$). As visualized in the figure, this boundary aligns with maximum $\hat{\alpha}$ implied by the PAC guarantee from Vovk [2012, Prop 2a], where differences are likely due to differences in bound derivation. However, there remains a gap between our analysis and the results from Vovk [2012, Prop 2b], suggesting room for future tighter bounds.

## 5.2 Optimizing efficiency via constrained stochastic gradient optimization

In order to select a $Q$ from this set, we propose directly optimizing the efficiency objective[3] (7). However, direct optimization is difficult due to (i) the expectation over $x$ and $\theta$, and (ii) the non-differentiability of the efficiency objective $\mathcal{L}_{\text{eff}}(C)$. To address (i), we use minibatches of inputs sampled from $D_{\text{cal}}$ and a finite set of $K$ samples of $\theta$ from $Q(\theta)$ to construct a Monte-Carlo approximation of the expectation, providing a stochastic estimate of the objective that can be used in a stochastic gradient descent algorithm. To ensure differentiability, we follow the strategy proposed in [Stutz et al., 2021] and replace any non-differentiable operations in the computation of $\mathcal{L}_{\text{eff}}$ with their "soft" differentiable counterparts. For example, the quantile operation needed to compute the optimal threshold $\tau(\theta, D_{\text{cal}}, \hat{\alpha})$ can be replaced with a soft quantile implementation leveraging a soft sorting

---

[3]Alternatively, one could optimize the generalization bound (9) directly, which effectively introduces a penalty on the KL divergence. In our case, we already have a constraint on the KL which mitigates overfitting, so we opt not to add complexity to the optimization.

---
**Algorithm 1** Optimal Conformal Prediction with Generalization Guarantees
---
**Require:** Parameterized score function $s(\boldsymbol{x}, \boldsymbol{y}, \theta)$, Calibration dataset $D_{\text{cal}}$, Target coverage rate $1 - \alpha$, Probability of correctness $\delta$, Differentiable efficiency objective $\ell_{\text{eff}}(C)$, Prior distribution $P(\theta) = \mathcal{N}(\theta; \mu_0, \Sigma_0)$, Posterior distribution family $Q(\theta) = \mathcal{N}(\theta; \mu, \Sigma)$, Empirical coverage $1 - \hat{\alpha}$
  Initialize posterior distribution parameters $\mu, \Sigma \leftarrow \mu_0, \Sigma_0$
  **for** each optimization iteration **do**
    $\hat{\mathcal{L}}_{\text{eff}}(\mu, \Sigma) \leftarrow \text{DIFFERENTIABLELOSS}(\mu, \Sigma, \alpha)$
    $c(\mu, \Sigma) \leftarrow \text{KL}(\mathcal{N}(\mu, \Sigma) \| \mathcal{N}(\mu_0, \Sigma_0)) - B(\alpha, \hat{\alpha}, \delta, N)$
    Update $\mu, \Sigma$ to minimize $\mathcal{L}_{\text{eff}}(\mu, \Sigma)$ subject to $c(\mu, \Sigma) < 0$
  **end for**
  **function** DIFFERENTIABLELOSS($\mu, \Sigma, \alpha$)
    Sample mini-batch $S = \{(\boldsymbol{x}_j, \boldsymbol{y}_j)\}_{j=1}^{J}$ from $D_{\text{cal}}$
    Sample $\zeta_1, \ldots, \zeta_K \sim \mathcal{N}(\mathbf{0} \in \mathbb{R}^d, I_d)$
    $\theta_k \leftarrow \mu + \Sigma \zeta_k$ for all $k \in \{1, \ldots, K\}$
    $s_{j,k} \leftarrow s(\boldsymbol{x}_j, \boldsymbol{y}_j; \theta_k)$ for all $j \in \{1, \ldots, J\}, k \in \{1, \ldots, K\}$
    $\tau_k \leftarrow \text{SOFTQUANTILE}(s_{[:,k]}, \lceil (J+1)(1 - \hat{\alpha}) \rceil / J)$ for all $k \in \{1, \ldots, K\}$
    Compute prediction sets $C_{j,k} = C(\boldsymbol{x}_j; \tau_k)$ for all $j \in \{1, \ldots, J\}, k \in \{1, \ldots, K\}$
    **return** $\hat{\mathcal{L}}_{\text{eff}}(\mu, \Sigma) = \frac{1}{JK} \sum_{k=1}^{K} \sum_{j=1}^{J} \ell_{\text{eff}}(C_{j,k})$
  **end function**
---

algorithm [Cuturi et al., 2019, Grover et al., 2019]. As optimizing over the space of all probability distributions over $\boldsymbol{\Theta}$ is intractable, we restrict $Q$ to be within a parametric class of distributions for which obtaining samples and evaluating the KL divergence with respect to the prior is tractable. In this work, we follow prior work in the PAC-Bayes and Bayesian Neural Network literature and fix both $P$ and $Q$ to be a Gaussian with a diagonal covariance matrix, $\mathcal{N}(\boldsymbol{\mu}, \text{diag}(\boldsymbol{\sigma}^2))$, where $\boldsymbol{\mu} \in \mathbb{R}^d$ and $\boldsymbol{\sigma}^2 \in \mathbb{R}_+^d$. This allows analytic evaluation of the KL divergence, and differentiable sampling via the reparametrization trick, as shown in Alg. 1. In this way, both the optimization objective and the constraint can be evaluated in a differentiable manner, and fed into a gradient based constrained optimization algorithm. We choose the $Q$ with the lowest loss that still satisfies Corollary 2.1. More details on the optimization procedure can be found in Appendix B.

## 5.3 Practical considerations

**Choice of prior.** A prior $P(\theta)$ may give high likelihood to values of $\theta$ that yield good prediction sets, but also to $\theta$ that perform poorly. To optimize efficiency, $Q$ must shift probability mass away from poor $\theta$ towards good $\theta$. However, due to the asymmetry of the KL divergence, $\text{KL}(Q\|P)$ is larger if $Q$ assigns probability mass where $P$ does not than vice-versa. Thus, in order to effectively optimize efficiency under a tight KL budget, we must choose $P$ that not only (i) assigns minimal probability density to bad values of $\theta$, but more importantly, (ii) assigns significant probability density to good choices of $\theta$. How do we choose an effective prior? In some cases, it suffices to choose $P$ as an isotropic Gaussian distribution centered around a random initialization of $\theta$. However, especially when $\theta$ correspond to parameters of a neural network, using a data-informed prior can lead to improved performance [Perez-Ortiz et al., 2021, Dziugaite and Roy, 2018]. Care is needed to ensure that a data-informed prior does not break the generalization guarantee; we find that a simple data-splitting approach similar to [Perez-Ortiz et al., 2021] is effective: We split $D_{\text{cal}}$ into two disjoint sets: $D_0$, used to optimize the prior without any constraints, and $D_N$, used to optimize the posterior according to Alg. 1. In our experiments, we consider two methods of optimizing a Gaussian prior $P = \mathcal{N}(\boldsymbol{\mu}, \text{diag}(\boldsymbol{\sigma}^2))$: First, we can optimize only the mean $\boldsymbol{\mu}$ to minimize $\mathcal{L}_{\text{eff}}(\boldsymbol{\mu})$, keeping $\boldsymbol{\sigma}^2$ fixed. This aims to shift $P$ to assign more density to good values of $\theta$, but the fixed $\sigma^2$ may still assign significant mass to bad choices of $\theta$. Alternatively, we consider optimizing both $\boldsymbol{\mu}$ and $\boldsymbol{\sigma}^2$ to minimize $\mathcal{L}_{\text{eff}}(P)$. By also optimizing the variance, we can minimize probability mass assigned to bad choices of $\theta$, but this comes with a risk of overfitting to $D_0$.

**Test-time evaluation.** The guarantees on coverage and efficiency generalization hold in expectation over both data sampled from the data distribution $\mathcal{D}$ as well as model and score function parameters sampled from the learned posterior $Q$. Thus, in order to use such a predictor in practice and attain the desired coverage and efficiency, one would need to sample parameters $\theta$ from $Q$, and then compute the

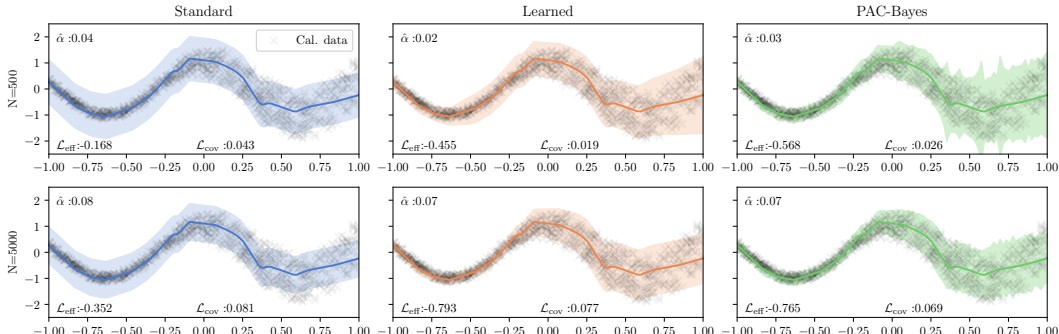

Figure 3: Standard ICP with a fixed score function (left) can yield over-conservative prediction sets. By using calibration data to also learn an uncertainty scaling factor in the nonconformity score function, both the learned (middle) and PAC-Bayes (right) approaches can yield more efficient prediction sets. When the calibration set size is large (bottom), both the learned and PAC-Bayes approaches do well. However, when calibration data is limited (top), our PAC-Bayes approach yields prediction sets with better test-time efficiency.

resulting prediction set $C(\boldsymbol{x}; \tau)$. However, as the threshold for this prediction set $\tau = \tau^*(D_{\mathrm{cal}}, \hat{\alpha}; \theta)$ depends on $D_{\mathrm{cal}}$, this approach would require holding on to the calibration dataset to find the optimal threshold of each sample of $\theta$. In practice, once we optimize $Q$, we pre-sample $K$ values of $\theta$ from $Q$, and pre-compute the optimal threshold for each. Then, at test-time, we randomly select one $\theta, \tau$ pair from the set to evaluate each test input.

## 6 Experimental Results

We evaluate our approach on an illustrative regression problem as well as on an MNIST classification scenario with simulated distributional shift.[4] For each domain, we evaluate our *PAC-Bayes* approach against two baselines: (i) the *standard* ICP approach, where we use a fixed, data-independent score function $s_{\mathrm{default}}(\boldsymbol{x}, \boldsymbol{y})$ and use the entirety of the calibration data to compute the critical threshold $\tau$; and (ii) a *learned* ICP approach [Stutz et al., 2021] which uses a portion of $D_{\mathrm{cal}}$ to optimize parameters $\theta$ of a parametric score function $s_{\mathrm{optim}}(\boldsymbol{x}, \boldsymbol{y}; \theta)$ to minimize an efficiency loss $\mathcal{L}_{\mathrm{eff}}$, and the remaining portion to re-estimate the threshold $\tau$ to guarantee test-time coverage. For the PAC-Bayes method, we consider the same parametric form as the learned baseline, but instead randomize the score function by modeling a distribution over $\theta$. Both the learned and the PAC-Bayes baseline require choosing a *data split* ratio, i.e. what fraction is used for optimization (or prior tuning for the PAC-Bayes approach), and what fraction is held out for recalibration (or constrained optimization). For all methods, we target the same test-time coverage guarantee of $1 - \alpha$ coverage with probability greater $1 - \delta$. For the standard and learned methods, the amount of data in the held-out recalibration set determines the empirical coverage level $\hat{\alpha}$ needed to attain this guarantee, using either Vovk Prop 2a (4), or Vovk Prop 2b (5). We compare against both bounds in our experiments. For the PAC-Bayes method, the choice of $\hat{\alpha}$ is a hyperparameter: choosing a lower value yields a larger budget for optimization, but entails using a more extreme quantile for the threshold $\tau$. As all methods provide guarantees on test-time coverage, our evaluation focuses on the efficiency of each method's predictions on held-out test data that is exchangeable with the calibration data. Additional results (including coverage results), as well as specific details for all experiments, can be found in App. B.

### 6.1 Illustrative demonstration: 1-D regression

As an illustrative example, we consider a 1-D regression problem where inputs $\boldsymbol{x} \in \mathbb{R}$ and targets $\boldsymbol{y} \in \mathbb{R}$ are drawn from a heteroskedastic distribution where noise increases with $\boldsymbol{x}$, as shown in Figure 3. We train a two hidden layer fully connected neural network to minimize the mean squared error on this data, obtaining a base predictor $f(\boldsymbol{x})$ that produces a single point estimate for $\boldsymbol{y}$. We aim

---

[4]We implemented our approach using PyTorch [Paszke et al., 2019] and Hydra [Yadan, 2019]; code to run all experiments is available at `https://github.com/NVlabs/pac-bayes-conformal-prediction`

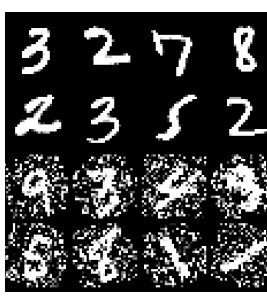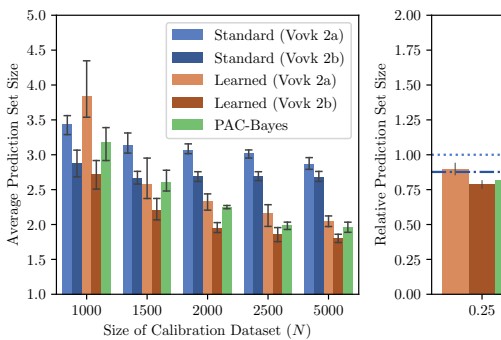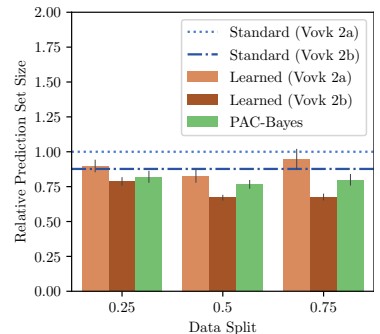

Figure 4: Left: A base model is trained on clean MNIST digits (top), but calibrated and tested on corrupted digits (bottom). Middle: Average prediction set size on test data versus calibration set size ($N$) for a data split ratio of $0.5$. Right: Prediction set size relative to standard ICP (Vovk 2a) as a function of the data split ratio, averaged over random seeds and across all calibration set sizes.

to obtain a set-valued predictor using a held-out set of calibration data with $\alpha = 0.1$ and $\delta = 0.05$. For the sake of this example, we assume that we no longer have access to the training data, and only have calibration data and a pre-trained base predictor which only outputs point estimates and no input-dependent estimate of variance. Therefore, for the standard application of ICP, we must use the typical nonconformity function for regression, $s_{\mathrm{default}}(\boldsymbol{x}, \boldsymbol{y}) = \|f(\boldsymbol{x}) - \boldsymbol{y}\|$. Note that this choice yields prediction sets with a fixed, input-independent size, which is suboptimal in this heteroskedastic setting. To address this, we consider using the calibration dataset not only to determine the threshold $\tau^*$, but also to *learn* a parametric score function that can capture this heteroskedasticity. Specifically, we learn an input-dependent uncertainty score $u(\boldsymbol{x}; \theta)$ which is used to scale the nonconformity score $s_{\mathrm{optim}}(\boldsymbol{x}, \boldsymbol{y}; \theta) = \|f(\boldsymbol{x}) - \boldsymbol{y}\| / u(\boldsymbol{x}; \theta)$ [Angelopoulos and Bates, 2021]. We model $u(\boldsymbol{x}, \theta)$ as a separate neural network with the same architecture as $f$, and optimize either $\theta$ or $Q(\theta)$ to minimize the volume of the resulting prediction sets. To do so, we minimize a loss proportional to the log of the radius of this set, which in this case can be written as $\ell_{\mathrm{eff}}(C(\boldsymbol{x}, \tau)) = \log(u(\boldsymbol{x}; \theta) \cdot \tau)$.

As can be seen in Figure 3, when $N = 5000$, both the learned baseline and our PAC-Bayes approach are able to learn an appropriate input-dependent uncertainty function which yields prediction sets which capture the heteroskedasticity of the data, and thus yield tighter sets in expectation over $\boldsymbol{x}$. However, if we reduce the amount of calibration data available to $N = 500$, the learned approach starts to overfit on the split used to optimize efficiency. In contrast, the KL-constraint of the PAC-Bayes approach mitigates this overfitting, and outperforms both baselines in terms of test set efficiency for smaller amount of calibration data.

## 6.2 Corrupted MNIST

Next, we consider a more challenging distribution shift scenario, where we aim to deploy a model trained on "clean" data to a domain with distribution shift. As the base predictor, we use a LeNet convolutional neural network trained on a softmax objective to classify noise-free MNIST digits [LeCun et al., 1998]. Then, we simulate distribution shift by adding Gaussian noise and applying random rotations to a held-out set of the dataset: examples of the clean and corrupted digits are shown in Figure 4. Our goal is to use a calibration dataset of this corrupted data to design a set-valued predictor that achieves coverage on held-out test data with the same corruption distribution.

As is conventional, when applying ICP to probabilistic classifiers, we use the negative log probability as the nonconformity score $s_{\mathrm{default}}(\boldsymbol{x}, \boldsymbol{y}) = [\log \mathrm{softmax}\, f(\boldsymbol{x})]_{[\boldsymbol{y}]}$ [Stutz et al., 2021, Angelopoulos and Bates, 2021]. This score function leverages the implicit uncertainty information encoded in the base model's probabilistic predictions; but these probabilities are far from calibrated, especially in the presence of distribution shift [Ovadia et al., 2019, Sharma et al., 2021]. Thus, we consider using the calibration dataset, which is representative of the corrupted test distribution, to fine-tune the predicted probabilities, defining $s_{\mathrm{optim}}(\boldsymbol{x}, \boldsymbol{y}, \theta) = [\log \mathrm{softmax}\, f(\boldsymbol{x}; \theta)]_{[\boldsymbol{y}]}$, where $\theta$ are the parameters of the fully connected layers of the base predictor. As before, we measure efficiency as the size of the prediction set, $\ell_{\mathrm{eff}}(C) = |C|$. In contrast to the regression setting, here $C$ is a discrete set over the set of class labels, and thus the size of this set is non-differentiable. To smooth this objective for training,

we follow the approach of [Stutz et al., 2021], and first use a sigmoid to compute soft set-membership assignment for each possible label, and then take the sum to estimate the size of the set,

$$\hat{\ell}_{\text{eff}}(C(\boldsymbol{x},\tau)) = \sum_{\boldsymbol{y} \in \{0,\ldots,9\}} \sigma(T^{-1}(\tau - s(\boldsymbol{x},\boldsymbol{y},\theta))), \tag{11}$$

where the temperature $T$ is a hyperparameter which controls the smoothness of the approximation; in our experiments we use $T = 0.1$.

We evaluate each method for different calibration set sizes $N$ and data split ratios, repeating each experiment using 3 random seeds. For all approaches, we desire a coverage guarantee of $\alpha = 0.1$ with $\delta = 0.05$. For the PAC-Bayes approach, we run the algorithm with $\delta = 0.01$, performing a grid search over 5 values of $\hat{\alpha}$ evenly spaced between $0.2$ and $0.8$, and choosing the approach with the best efficiency generalization certificate (9); using the union bound for probability ensures that the efficiency for the best of the five runs will hold with probability at least $1 - 0.05$, i.e., an effective $\delta$ of $0.05$. For the PAC-Bayes approach, we optimize both the prior mean $\boldsymbol{\mu}$ and variance $\boldsymbol{\sigma}^2$ on $D_0$, as we found that optimizing the mean alone yielded a poor prior; ablations are provided in App. B. The results are summarized in Figure 4. In the middle figure, we hold the data split fixed at $0.5$, and plot the average test set size as a function of $N$. First, we note that even for the standard ICP approach, increasing $N$ leads to a smaller prediction set size, as we are able to calibrate to a larger $\hat{\alpha}$ while still maintaining the desired test-time guarantee. In general, both the learned and PAC-Bayes approaches improve upon the standard baseline by fine-tuning on the corrupted calibration data. However, when data is limited ($N = 1000$), the learned baseline overfits during optimization on the first half of the calibration data, and subsequent calibration yields set sizes that are even larger than the standard method. Our method mitigates this by using all the data to simultaneously fine-tune and calibrate while ensuring test-time generalization, thanks to the KL constraint. Empirically, the PAC-Bayes method yields prediction sets with efficiency comparable or higher than the learned baseline calibrated using Vovk 2a. However, there still remains a gap between our results and the learned baseline calibrated with tighter bound in Vovk 2b, as is evident in our analysis of our bound in Figure 2. In the right figure, we explore the sensitivity of both learned methods on the choice of data split ratio. We observe that the PAC-Bayes method is less sensitive to the choice of data split ratio.

## 7 Conclusion

In this paper, we introduced theory for optimizing conformal predictors while retaining PAC generalization guarantees on coverage and efficiency *without* the need to hold-out data for calibration. We achieved this by combining PAC-Bayes theory with ICP to furnish generalization bounds with the *same* data used for training. We translated our theory into a practical algorithm and demonstrated its efficacy on both regression and classification problems.

**Limitations.** As shown in Figure 2, our theory remains overconservative relative to the tight Vovk 2b PAC bound, which future work could aim to address. Practically, a key limitation of our approach is its dependence on the availability of a good prior. Although, we were able to mitigate this issue by training a prior on a portion of the calibration dataset, this strategy can struggle when datasets are small or the number of parameters to optimize are very large. Furthermore, while diagonal Gaussian distributions are convenient to optimize, they may not represent the optimal posterior. These limitations may explain why we obtained only modest improvements in efficiency with respect to the Vovk 2a baselines in our experiments; we are optimistic that future advances in prior selection and representation may lead to improvements. Finally, while PAC-Bayes theory requires sampling from the posterior $Q$, our set-up also requires computing the threshold $\tau^*$ for each parameter sample. Future work could explore applying disintegrated PAC-Bayes theory [Viallard et al., 2023] to yield (perhaps more conservative) generalization bounds that hold for a single sample from the posterior.

**Broader Impact.** This work is building towards safer and trustworthy ML systems that can reason about uncertainty in their predictions. In particular, this work is a step towards developing calibrated uncertainty set predictors that are efficient enough for practical applications.

**Future work.** This work opens up exciting new theoretical and practical questions to pursue. On the theoretical front, we are excited to explore the development of an online learning approach that can leverage this theory to provably adapt to distribution shifts on the fly. On the practical front, we look forward to using this framework for providing reasonable uncertainty estimates in robot autonomy stacks to facilitate decision making and safety certification.

## Acknowledgments and Disclosure of Funding

Anirudha Majumdar was partially supported by the NSF CAREER Award [#2044149] and the Office of Naval Research [N00014-23-1-2148]. Asher Hancock was supported by the National Science Foundation Graduate Research Fellowship Program under Grant No. DGE-2146755. Any opinions, findings, and conclusions or recommendations expressed in this material are those of the author(s) and do not necessarily reflect the views of the National Science Foundation.

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

# A  Proofs

Our coverage and efficiency generalization bounds follow a similar form to many PAC-Bayes bounds. A fundamental building block of these proofs is the Donsker-Varadhan variational formula [Donsker and Varadhan, 1983], which we restate below.

**Lemma 1** (Donsker-Varadhan variational formula). *For any measurable bounded function $h : \Theta \to \mathbb{R}$, we have,*

$$\log \mathbb{E}_{\theta \sim P} [\exp(h(\theta))] = \sup_{Q \in \mathcal{P}(\theta)} \left[ \mathbb{E}_{\theta \sim Q} [h(\theta)] - \mathrm{KL}(Q||P) \right].$$

We then apply this lemma to obtain a core result relating to quantities depending on parameters $\theta$ and sampled data $D_N$, which we derive first as a second lemma which is common to the proof of both generalization bounds.

**Lemma 2.** *Let $\Phi(\theta, D_N) : \Theta \times (\mathcal{X} \times \mathcal{Y})^N \to \mathbb{R}$ be a measurable function mapping a parameter and a set of sampled data to a scalar. Let $P$ be an arbitrary distribution over $\Theta$, independent of $D_N$. Let $Q$ denote another arbitrary probability distribution over $\Theta$. Suppose there exists a function $B(N) > 0$ such that, for any fixed $\theta \in \Theta$,*

$$\mathbb{E}_{D_N} [\exp(\Phi(\theta, D_N))] \leq B(N).$$

*Then, we have that*

$$\mathbb{P}_{D_N} \left( \forall Q : \mathbb{E}_{\theta \sim Q} [\Phi(\theta, D_N)] \leq \mathrm{KL}(Q||P) + \log \left( \frac{B(N)}{\delta} \right) \right) \geq 1 - \delta.$$

*Proof of Lemma 2.* We start with our assumed property of $B(N)$, and integrate it with respect to $\theta$.

$$\mathbb{E}_{D_N} [\exp(\Phi(\theta, D_N))] \leq B(N)$$

$$\mathbb{E}_{\theta \sim P} \mathbb{E}_{D_N} [\exp(\Phi(\theta, D_N))] \leq B(N)$$

The integrand is non-negative and $P$ is independent of $D_N$; so, by Tonelli's theorem, we can swap the order of the integration with respect to the parameters $\theta$ and the dataset $D_N$, yielding

$$\mathbb{E}_{D_N} \mathbb{E}_{\theta \sim P} [\exp(\Phi(\theta, D_N))] \leq B(N)$$

Now, we can apply the Donsker-Varadhan variational formula to get

$$\mathbb{E}_{D_N} \left[ e^{\sup_{Q \in \mathcal{P}(\theta)} [\mathbb{E}_{\theta \sim Q} [\Phi(\theta, D_N)] - \mathrm{KL}(Q||P)]} \right] \leq B(N) \tag{12}$$

$$\mathbb{E}_{D_N} \left[ e^{\sup_{Q \in \mathcal{P}(\theta)} [\mathbb{E}_{\theta \sim Q} [\Phi(\theta, D_N)] - \mathrm{KL}(Q||P) - \log B(N)]} \right] \leq 1. \tag{13}$$

Now, by the Chernoff bound, for any random variable $X$,

$$\mathbb{P} (X > s) \leq \mathbb{E}[e^X] e^{-s},$$

so if we let $\delta = e^{-s}$, $s = -\log \delta$, we have

$$\mathbb{P} (X > -\log \delta) \leq \mathbb{E}[e^X] \delta.$$

Now, applying this bound to $X = \sup_{Q \in \mathcal{P}(\theta)} [\mathbb{E}_{\theta \sim Q} [\Phi(\theta, D_N)] - \mathrm{KL}(Q||P) - \log B(N)]$, we obtain

$$\mathbb{P}_{D_N} \left( \sup_{Q \in \mathcal{P}(\theta)} \left[ \mathbb{E}_{\theta \sim Q} [\Phi(\theta, D_N)] - \mathrm{KL}(Q||P) - \log B(N) \right] > -\log \delta \right)$$

$$\leq \mathbb{E}_{D_N} \left[ e^{\sup_{Q \in \mathcal{P}(\theta)} [\mathbb{E}_{\theta \sim Q} [\Phi(\theta, D_N)] - \mathrm{KL}(Q||P) - \log B(N)]} \right] \delta$$

$$\leq \delta \tag{14}$$

Now, since

$$\left(\exists Q \in \mathcal{P}(\theta) : \mathop{\mathbb{E}}_{\theta \sim Q}[\Phi(\theta, D_N)] - \mathrm{KL}(Q\|P) - \log B(N) > -\log \delta\right)$$

$$\implies \left(\sup_{Q \in \mathcal{P}(\theta)} \left[\mathop{\mathbb{E}}_{\theta \sim Q}[\Phi(\theta, D_N)] - \mathrm{KL}(Q\|P) - \log B(N)\right] > -\log \delta\right),$$

the result from (14) implies that

$$\mathop{\mathbb{P}}_{D_N}\left(\exists Q \in \mathcal{P}(\theta) : \left[\mathop{\mathbb{E}}_{\theta \sim Q}[\Phi(\theta, D_N)] - \mathrm{KL}(Q\|P) - \log B(N)\right] \leq -\log \delta\right) \leq \delta$$

Taking the complement, we have

$$\mathop{\mathbb{P}}_{D_N}\left(\forall Q \in \mathcal{P}(\theta) : \left[\mathop{\mathbb{E}}_{\theta \sim Q}[\Phi(\theta, D_N)] - \mathrm{KL}(Q\|P) - \log B(N)\right] \leq -\log \delta\right) \geq 1 - \delta$$

Rearranging terms completes the proof. $\qquad\square$

Equipped with this lemma, we can now prove our core theorems. We start by proving the coverage bound.

*Proof of Theorem 1.* First, let $\mathcal{L}_{\mathrm{cov}}(\theta, D_N, \hat{\alpha})$ represent the miscoverage rate of a conformal predictor calibrated to achieve $1 - \hat{\alpha}$ empirical coverage on $D_N$:

$$\mathcal{L}_{\mathrm{cov}}(\theta, D_N, \hat{\alpha}) = \mathop{\mathbb{P}}_{\boldsymbol{x}, \boldsymbol{y} \sim \mathcal{D}}\left(\boldsymbol{y} \notin C(\boldsymbol{x}; \tau^*(D_N, \hat{\alpha}; \theta), \theta)\right).$$

Note that for a fixed $\theta$, the distribution of coverage (and hence, mis-coverage) conditioned on a particular calibration dataset follows a Beta distribution [Angelopoulos and Bates, 2021, Vovk, 2012]. Specifically, $\mathcal{L}_{\mathrm{cov}}(\theta, D_N, \hat{\alpha}) \sim \mathrm{Beta}((\lfloor(N+1)\hat{\alpha}\rfloor, N + 1 - \lfloor(N+1)\hat{\alpha}\rfloor))$.

Now, choose $\Phi(\theta, D_N)$ to be

$$\Phi(\theta, D_N) = (N - 1)\mathrm{kl}\left(\frac{\lfloor(N+1)\hat{\alpha}\rfloor - 1}{N - 1} \,\middle\|\, \mathcal{L}_{\mathrm{cov}}(\theta, D_N, \hat{\alpha})\right).$$

First, we derive the bound $B(N)$ required to apply Lemma 2. Let $k = \lfloor(N+1)\hat{\alpha}\rfloor$, $a = \frac{k-1}{N-1}$, and $b = \mathcal{L}_{\mathrm{cov}}(\theta, D_N, \hat{\alpha})$. Then,

$$\mathop{\mathbb{E}}_{D_N}[\exp(\Phi(\theta, D_N))]$$

$$= \mathop{\mathbb{E}}_{D_N}[\exp((N-1)\mathrm{kl}(a\|b))]$$

$$= \mathop{\mathbb{E}}_{D_N}\left[\exp\left((N-1)a\log\frac{a}{b} + (N-1)(1-a)\log\frac{1-a}{1-b}\right)\right]$$

$$= \mathop{\mathbb{E}}_{D_N}\left[\left(\frac{a}{b}\right)^{(N-1)a}\left(\frac{1-a}{1-b}\right)^{(N-1)(1-a)}\right]$$

Note that $(N-1)a = k - 1$, and $(N-1)(1-a) = N - k$. Substituting, we have

$$\mathop{\mathbb{E}}_{D_N}[\exp(\Phi(\theta, D_N))] = \mathop{\mathbb{E}}_{D_N}\left[\left(\frac{a}{b}\right)^{k-1}\left(\frac{1-a}{1-b}\right)^{N-k}\right]$$

$$= a^{k-1}(1-a)^{N-k}\mathop{\mathbb{E}}_{D_N}\left[b^{1-k}(1-b)^{k-N}\right]$$

$$= a^{k-1}(1-a)^{N-k}\int_0^1 b^{1-k}(1-b)^{k-N}\mathrm{Beta}(b; k, N+1-k)db$$

$$= a^{k-1}(1-a)^{N-k}\int_0^1 b^{1-k}(1-b)^{k-N}\frac{b^{k-1}(1-b)^{N-k}N!}{(k-1)!(N-k)!}db$$

$$= a^{k-1}(1-a)^{N-k}\frac{N!}{(k-1)!(N-k)!}$$

$$= \mathrm{Beta}(a; k, N+1-k)$$

Thus, in this case we can analytically evaluate this expectation as

$$\underset{D_N}{\mathbb{E}}\left[\exp(\Phi(\theta, D_N))\right] = B(N) := \text{Beta}((k-1)/(N-1); k, N+1-k). \tag{15}$$

where $\text{Beta}(x; \alpha, \beta)$ is the pdf of the Beta distribution with parameters $\alpha, \beta$.

To compare our bound with other PAC-Bayes bounds and to quantify the asymptotic behavior of $B(N)$, we can derive an upper bound. First, we plug in our definition of $a$, and group terms

$$B(N) = a^{k-1}(1-a)^{N-k}\frac{N!}{(k-1)!(N-k)!} = N\frac{(N-1)!}{(N-1)^{N-1}}\frac{(k-1)^{k-1}}{(k-1)!}\frac{(N-k)^{N-k}}{(N-k)!}$$

By Stirling's approximation of $a!$, we know

$$\sqrt{2\pi a}a^a e^{-a+\frac{1}{12a+1}} < a! < \sqrt{2\pi a}a^a e^{-a+\frac{1}{12a}}$$

Therefore,

$$\frac{a^a}{a!} < \frac{1}{\sqrt{2\pi a}}e^{a-\frac{1}{12a+1}}$$

$$\frac{a!}{a^a} < \sqrt{2\pi a}e^{-a+\frac{1}{12a}}$$

Plugging in these bounds, we can obtain a simplified expression which removes the factorials and high powers of $N$:

$$B(N) < N\frac{\sqrt{(N-1)}}{\sqrt{2\pi(k-1)(N-k)}}\exp\left(\frac{1}{12N-12} - \frac{1}{12k-11} - \frac{1}{12N-12k+1}\right)$$

Notice that for $0 < k < N$, the term in the exponent is negative. Given the definition of $k$, this is true as long as $N \geq \frac{1}{\hat{\alpha}} - 1$. In this case, due to monotonicity, the exponential term is less than 1. This yields

$$B(N) < N\frac{\sqrt{(N-1)}}{\sqrt{2\pi(k-1)(N-k)}} := O\left(\sqrt{\frac{N}{\hat{\alpha}(1-\hat{\alpha})}}\right). \tag{16}$$

Having computed $B(N)$, we can now apply Lemma 2 to obtain

$$\underset{D_N}{\mathbb{P}}\left(\forall Q: \underset{\theta \sim Q}{\mathbb{E}}\left[(N-1)\text{kl}\left(\frac{\lfloor(N+1)\hat{\alpha}\rfloor - 1}{N-1} \,\middle\|\, \mathcal{L}_{\text{cov}}(\theta, D_N, \hat{\alpha})\right)\right] \leq \text{KL}(Q\|P) + \log\left(\frac{B(N)}{\delta}\right)\right) \geq 1 - \delta.$$

Since $\text{kl}(\cdot, \cdot)$ is convex in both arguments, we can use Jensen's inequality to bring the expectation over $\theta$ inside each argument, yielding,

$$\underset{D_N}{\mathbb{P}}\left(\forall Q: (N-1)\text{kl}\left(\frac{\lfloor(N+1)\hat{\alpha}\rfloor - 1}{N-1} \,\middle\|\, \underset{\theta \sim Q}{\mathbb{E}}\left[\mathcal{L}_{\text{cov}}(\theta, D_N, \hat{\alpha})\right]\right) \leq \text{KL}(Q\|P) + \log\left(\frac{B(N)}{\delta}\right)\right) \geq 1 - \delta.$$

Note that by definition in equation (6), $\mathcal{L}_{\text{cov}}(Q) = \mathbb{E}_{\theta \sim Q}\left[\mathcal{L}_{\text{cov}}(\theta, D_N, \hat{\alpha})\right]$. Rearranging terms completes the proof. $\qquad\square$

*Proof of Corollary 2.1.* Let $q = \frac{\lfloor(n+1)\hat{\alpha}\rfloor - 1}{n-1}$, and for compactness, let $\mathcal{L}_{\text{cov}} = \mathcal{L}_{\text{cov}}(\theta, \tau(\theta, D_N, \hat{\alpha}))$ Let $C$ be the event that our KL constraint holds, i.e.

$$\text{KL}(Q\|P) \leq (n-1)\text{kl}(q\|\alpha) - \log\left(\frac{B(n)}{\delta}\right)$$

Consider the event $E$ that $\mathbb{E}_{\theta \sim Q}[\mathcal{L}_{\text{cov}}] \leq q$. Since $q \leq \alpha$, we have that

$$\mathbb{P}\left(\underset{\theta \sim Q}{\mathbb{E}}[\mathcal{L}_{\text{cov}}] \leq \alpha \,\middle|\, E, C\right) = 1.$$

Now, consider the complement, $\bar{E}$. We know that $\text{kl}(q\|x)$ is monotonically increasing in $x$ for $x \geq q$. Thus, conditioned on $\bar{E}$, we have

$$\mathbb{P}\left(\underset{\theta \sim Q}{\mathbb{E}}[\mathcal{L}_{\text{cov}}] \leq \alpha \,\middle|\, \bar{E}, C\right) = \mathbb{P}\left(\text{kl}\left(q \,\middle\|\, \underset{\theta \sim Q}{\mathbb{E}}[\mathcal{L}_{\text{cov}}]\right) \leq \text{kl}(q \,\|\, \alpha) \,\middle|\, \bar{E}\right).$$

Given these statements, we have

$$
\mathbb{P}\left(\underset{\theta\sim Q}{\mathbb{E}}[\mathcal{L}_{\mathrm{cov}}]\leq\alpha\Big|C\right)
$$

$$
= \mathbb{P}\left(E\mid C\right)\mathbb{P}\left(\underset{\theta\sim Q}{\mathbb{E}}[\mathcal{L}_{\mathrm{cov}}]\leq\alpha\Big|E,C\right) + \mathbb{P}\left(\bar{E}\mid C\right)\mathbb{P}\left(\underset{\theta\sim Q}{\mathbb{E}}[\mathcal{L}_{\mathrm{cov}}]<\alpha\Big|\bar{E},C\right)
$$

$$
= *\,\mathbb{P}\left(E\mid C\right) + \mathbb{P}\left(\bar{E}\mid C\right)\mathbb{P}\left(\mathrm{kl}\left(q\,\|\,\underset{\theta\sim Q}{\mathbb{E}}[\mathcal{L}_{\mathrm{cov}}]\right)\leq\mathrm{kl}\left(q\,\|\,\alpha\right)\Big|\bar{E},C\right)
$$

$$
\geq \mathbb{P}\left(E\mid C\right)\mathbb{P}\left(\mathrm{kl}\left(q\,\|\,\underset{\theta\sim Q}{\mathbb{E}}[\mathcal{L}_{\mathrm{cov}}]\right)\leq\mathrm{kl}\left(q\,\|\,\alpha\right)\Big|E,C\right)
$$

$$
+ \mathbb{P}\left(\bar{E}\mid C\right)\mathbb{P}\left(\mathrm{kl}\left(q\,\|\,\underset{\theta\sim Q}{\mathbb{E}}[\mathcal{L}_{\mathrm{cov}}]\right)\leq\mathrm{kl}\left(q\,\|\,\alpha\right)\Big|\bar{E},C\right)
$$

$$
= \mathbb{P}\left(\mathrm{kl}\left(q\,\|\,\underset{\theta\sim Q}{\mathbb{E}}[\mathcal{L}_{\mathrm{cov}}]\right)\leq\mathrm{kl}\left(q\,\|\,\alpha\right)\Big|C\right) \tag{17}
$$

Now, due to the convexity of kl, we have through Jensen's inequality,

$$
\underset{\theta\sim Q}{\mathbb{E}}\left[\mathrm{kl}\left(q\|\mathcal{L}_{\mathrm{cov}}\right)\right]\geq\mathrm{kl}\left(q\,\|\,\underset{\theta\sim Q}{\mathbb{E}}[\mathcal{L}_{\mathrm{cov}}]\right).
$$

Plugging this in to the result of Theorem 1, we have

$$
\mathbb{P}\left(\forall Q,\mathrm{kl}\left(q\|\underset{\theta\sim Q}{\mathbb{E}}[\mathcal{L}_{\mathrm{cov}}]\right)\leq\frac{\mathrm{KL}(Q\|P)+\log\left(\frac{B(n)}{\delta}\right)}{n-1}\right)\geq 1-\delta
$$

Now, since this holds for all $Q$, it must also hold for those $Q$ satisfying the constraint. Thus,

$$
\mathbb{P}\left(\forall Q,\mathrm{kl}\left(q\|\underset{\theta\sim Q}{\mathbb{E}}[\mathcal{L}_{\mathrm{cov}}]\right)\leq\frac{\mathrm{KL}(Q\|P)+\log\left(\frac{B(n)}{\delta}\right)}{n-1}\Big|C\right)\geq 1-\delta
$$

$$
\mathbb{P}\left(\forall Q,\mathrm{kl}\left(q\|\underset{\theta\sim Q}{\mathbb{E}}[\mathcal{L}_{\mathrm{cov}}]\right)\leq\frac{(n-1)\mathrm{kl}\left(q\|\alpha\right)-\log\left(\frac{B(n)}{\delta}\right)+\log\left(\frac{B(n)}{\delta}\right)}{n-1}\Big|C\right)\geq 1-\delta
$$

$$
\mathbb{P}\left(\forall Q,\mathrm{kl}\left(q\|\underset{\theta\sim Q}{\mathbb{E}}[\mathcal{L}_{\mathrm{cov}}]\right)\leq\mathrm{kl}\left(q\|\alpha\right)\Big|C\right)\geq 1-\delta
$$

Plugging this result into (17), we have

$$
\mathbb{P}\left(\underset{\theta\sim Q}{\mathbb{E}}[\mathcal{L}_{\mathrm{cov}}]\leq\alpha\Big|C\right)\geq 1-\delta,
$$

completing the proof. $\qquad\square$

Next, we prove our bound on efficiency.

*Proof of Theorem 2.* We wish to bound $\mathcal{L}_{\mathrm{eff}}(Q)$ as a function of the observed empirical efficiency. Define $\hat{\mathcal{L}}_{\mathrm{eff}}(\theta,\tau,D_N)$ as the empirical mean efficiency observed for a particular value of parameter $\theta$ and threshold $\tau$, and $\mathcal{L}_{\mathrm{eff}}(\theta,\tau)$ as the expected efficiency of the prediction sets constructed with $\theta$ and $\tau$ on new data sampled from $\mathcal{D}$:

$$
\hat{\mathcal{L}}_{\mathrm{eff}}(\theta,\tau,D_N):=\frac{1}{N}\sum_{i=1}^{N}\ell_{\mathrm{eff}}(C(\boldsymbol{x}_i;\tau,\theta))
$$

$$
\mathcal{L}_{\mathrm{eff}}(\theta,\tau):=\underset{\boldsymbol{x}\sim\mathcal{D}}{\mathbb{E}}\left[\ell_{\mathrm{eff}}(C(\boldsymbol{x};\tau,\theta))\right].
$$

In ICP, the threshold $\tau = \tau^*(D_N, \hat{\alpha}, \theta)$ itself depends on $D_N$. This means that even for a fixed $\theta$, the observed efficiency for the prediction set on each sample in $D_N$ are not independent from one another. To avoid this complication, we define the quantity

$$\hat{R}(\theta, D_N) := \sup_\tau \|\hat{\mathcal{L}}_{\text{eff}}(\theta, \tau, D_N) - \mathcal{L}_{\text{eff}}(\theta, \tau)\|,$$

which measures the worst-case difference between the empirical and true mean efficiencies over the worst-case choice of threshold. Note that

$$\mathcal{L}_{\text{eff}}(\theta, \tau^*(D_N, \hat{\alpha}, \theta)) - \hat{\mathcal{L}}_{\text{eff}}(\theta, \tau^*(D_N, \hat{\alpha}, \theta), D_N) \leq \hat{R}(\theta, D_N). \tag{18}$$

Next, we consider the expectation of this quantity over the sampling of $D_N$:

$$R(\theta) := \mathbb{E}_{D_N}\left[\sup_\tau \|\hat{\mathcal{L}}_{\text{eff}}(\theta, \tau, D_N) - \mathcal{L}_{\text{eff}}(\theta, \tau)\|\right]$$

Given that, by assumption, $\ell_{\text{eff}}$ is $L_\tau$-Lipschitz in $\tau$, and $\tau$ is bounded by $\beta$ (due to $s(\boldsymbol{x}, \boldsymbol{y}, \theta)$ being bounded), we use the result shown in Proposition 4 of Bai et al. [2022] to show that $R(\theta)$ can be bounded for a fixed $\theta$:

$$R(\theta) \leq \frac{2\beta L_\tau}{\sqrt{N}}. \tag{19}$$

Furthermore, assuming $\ell_{\text{eff}}$ is bounded by $[0, 1]$, then, for a fixed $\theta$, $\hat{R}(\theta, D_N)$ satisfies the bounded differences property with bound $1/N$. Therefore, applying McDairmid's inequality, we obtain

$$\mathbb{P}_{D_N}\left(|\hat{R}(\theta, D_N) - R(\theta)| \geq s\right) \leq 2\exp\left(-2Ns^2)\right) \tag{20}$$

This bound only holds for a fixed $\theta$ independent of $D_N$.

To obtain a bound which holds for $\theta \sim Q$ for all $Q$, we can follow the standard PAC-Bayes bound sub-Gaussian random variables. Specifically, we turn to Lemma 2, this time defining

$$\Phi(\theta, D_N) = 2(N-1)(\hat{R}(\theta, D_N) - R(\theta))^2$$

Note that given (20), applying Lemma 1.5 , we can bound

$$\mathbb{E}_{D_N}\left[\exp(\Phi(\theta, D_N))\right] \leq B(N) := 2N.$$

Now, we can apply Lemma 2 to obtain

$$\mathbb{P}_{D_N}\left(\forall Q : \mathbb{E}_{\theta \sim Q}\left[2(N-1)(\hat{R}(\theta, D_N) - R(\theta))^2\right] \leq \text{KL}(Q\|P) + \log\left(\frac{2N}{\gamma}\right)\right) \geq 1 - \gamma$$

Bringing the expectation into the quadratic using Jensen's inequality, and rearranging terms, we obtain

$$\mathbb{P}_{D_N}\left(\forall Q : \mathbb{E}_{\theta \sim Q}\left[\hat{R}(\theta, D_N)\right] \leq \mathbb{E}_{\theta \sim Q}[R(\theta)] + \sqrt{\frac{\text{KL}(Q\|P) + \log\left(\frac{2N}{\gamma}\right)}{2(N-1)}}\right) \geq 1 - \gamma$$

Now, substituting (19) and (18), we obtain that with probability greater than $1 - \gamma$, for all $Q$, it holds that

$$\mathbb{E}_{\theta \sim Q}\left[\mathcal{L}_{\text{eff}}(\theta, \tau^*(D_N, \hat{\alpha}, \theta)) - \hat{\mathcal{L}}_{\text{eff}}(\theta, \tau^*(D_N, \hat{\alpha}, \theta), D_N)\right] \leq \frac{2\beta L_\tau}{\sqrt{N}} + \sqrt{\frac{\text{KL}(Q\|P) + \log\left(\frac{2N}{\gamma}\right)}{2(N-1)}}$$

$$\mathbb{E}_{\theta \sim Q}\left[\mathcal{L}_{\text{eff}}(\theta, \tau^*(D_N, \hat{\alpha}, \theta))\right] \leq \mathbb{E}_{\theta \sim Q}\left[\hat{\mathcal{L}}_{\text{eff}}(\theta, \tau^*(D_N, \hat{\alpha}, \theta), D_N)\right]$$

$$+ \frac{2\beta L_\tau}{\sqrt{N}} + \frac{\sqrt{\frac{1}{2}\text{KL}(Q\|P) + \frac{1}{2}\log\left(\frac{2N}{\gamma}\right)}}{\sqrt{N-1}}$$

Noting that by definition, $\mathcal{L}_{\text{eff}}(Q) = \mathbb{E}_{\theta \sim Q}\left[\mathcal{L}_{\text{eff}}(\theta, \tau^*(D_N, \hat{\alpha}, \theta))\right]$, which yields the theorem as stated, concluding the proof. $\square$

# B  Implementation Details

All experiments were performed on a workstation with a Intel Core i9-10980XE CPU with a NVIDIA GeForce RTX 3090 GPU.

## B.1  Detailed Algorithm Overview

In this section, we give a detailed description of our general experimental method. Subsequent sections give specifics to each of the experimental domains we consider.

**Model training.**  For each experiment, we first train a base neural network model on a training dataset. Next, for each calibration approach, we pair the neural network model with a score function which is used for calibration.

**Model calibration.**  In the calibration phase, we first split the calibration data $D_{\mathrm{cal}}$ into two random splits, a tuning dataset $D_0$ and true calibration dataset $D_N$, where the fraction of data used for tuning (the *data split*) is a hyperparameter. In the *standard* baseline, the no optimization is performed so $D_0$ is the empty set, and $D_N = D_{\mathrm{cal}}$ is used to calibrate the model. In the *learned* baseline, parameters are optimized on only $D_0$, and $D_N$ is used to calibrate by computing the threshold $\tau^*$. In our *PAC-Bayes* approach, we tune prior parameters on $D_0$, and then optimize the posterior following Algorithm 1 on $D_N$.

We aim for the same PAC guarantee of $1 - \alpha$ test coverage rate with probability greater than $1 - \delta$. For the *standard* and *learned* baselines, $N$, i.e. the size of $D_N$, determines the required empirical miscoverage rate $\hat{\alpha}$. We can either use $\hat{\alpha}$ equal to the value given by (4) [Vovk, 2012, Prop 2a], or by finding the value (e.g. by grid search) satisfying (5) [Vovk, 2012, Prop 2b]. For our *PAC-Bayes* approach, $\hat{\alpha}$ is a hyperparameter which trades off conservatism in choosing $\tau^*$ for the freedom to optimize score function parameters on $D_N$ while retaining the PAC generalization guarantees. To select this hyperparameter, we choose $K$ values for $\hat{\alpha}$, and for each, run our algorithm with a tighter $\delta' = \delta/K$. This allows us to use a union bound argument to achieve the same PAC guarantee as the other approaches *uniformly* over the $K$ predictors we obtain. We select between these predictors by choosing the predictor with the best generalization bound on efficiency (9).

**Optimization on $D_0$.**  In both the *learned* and *PAC-Bayes* approaches, we optimize parameters on the tuning dataset to minimize the efficiency via gradient descent. Note that the efficiency of a conformal predictor depends on the score function evaluated at a given input $x$, as well as the threshold $\tau^*$ which corresponds to the empirical quantile of the score function evaluated on the calibration dataset. We follow the approach of Stutz et al. [2021], and for each minibatch, we first compute $\tau^*$ by evaluating the $\hat{\alpha}$ quantile on the minibatch, and then use this threshold to evaluate the efficiency loss on the data. We use the softsort approach proposed by [Grover et al., 2019] to differentiate through the quantile function. The prior tuning in our *PAC-Bayes* approach follows the same protocol, with the addition of first sampling $\theta \sim \mathcal{N}(\mu_0, \Sigma_0)$ (differentiably, via the reparameterization trick), before computing the efficiency loss for each sample and averaging.

**Optimization on $D_N$.**  In our *PAC-Bayes* approach, we further optimize the score function on the second split of data, this time holding the prior fixed and instead optimizing the posterior $Q = \mathcal{N}(\mu, \Sigma)$ to minimize the efficiency loss (7) subject to the KL constraint (10). The efficiency loss is evaluated in the same way as above, and the KL constraint can be evaluated analytically. To perform the constrained optimization, we implement an augmented Lagrangian method. Specifically, we construct an unconstrained optimization problem with the objective

$$\mathcal{L}_{\mathrm{aug}}(\mu, \Sigma, s; \lambda, \rho) = \hat{\mathcal{L}}_{\mathrm{eff}}(\mu, \Sigma) + \lambda c(\mu, \Sigma, s) + \frac{\rho}{2} c(\mu, \Sigma, s)^2$$

$$c(\mu, \Sigma, s) = \mathrm{KL}(\mathcal{N}(\mu_0, \Sigma_0) || \mathcal{N}(\mu, \Sigma)) - B(\alpha, \hat{\alpha}, \delta, N) + s$$

where $B(\alpha, \hat{\alpha}, \delta, N)$ is the KL budget defined in (10) and $s \in \mathbb{R}_+$ is a positive slack variable translating the inequality constraint in (10) to an equality constraint $c(\mu, \Sigma, s) = 0$. In each outer iteration, we approximately solve this unconstrained optimization problem for $\mu, \Sigma$ and $s$ using projected gradient descent (clamping $s$ to 0 after any gradient step making it negative). After

optimization, we evaluate the constraint $c(\mu, \Sigma, s)$, and update the penalty scale according to

$$\lambda \leftarrow \lambda + \rho c(\mu, \Sigma, s)$$

before rerunning the inner optimization. We run 10 outer iterations, and return the $\mu, \Sigma$ which yielded the best efficiency loss while satisfying the constraint.

**Calibration on $D_N$.**  For all models, we calibrate using $D_N$ by choosing $\tau^*(D_N, \hat{alpha})$ to be the $q = \lceil (n+1)(1-\hat{\alpha}) \rceil / n$ quantile of the set of score function values evaluated on $D_N$. In the case of the *PAC-Bayes* approach, the score function (and hence the threshold) depends on the sampled value of $\theta$. Therefore, after optimization, we first pre-sample $M$ values of $\theta \sim Q$, and for each, compute $\tau^*$ as above. We store the sampled $\theta$ values along with their associated $\tau^*$ values for use at test time.

**Evaluation.**  To evaluate we construct prediction sets according to the learned score functions on a held-out test set of $N_{\text{test}}$ points. For each point, the prediction set is constructed according to (2) using the $\theta, \tau^*$ pair obtained from the calibration phase. In the case of the *PAC-Bayes* baseline, each test sample is randomly assigned to one of the pre-sampled parameter/threshold pairs. We measure the rate of coverage as well as the efficiency $\ell_{\text{eff}}$ of each prediction set. We repeat the entire process from calibration and evaluation on multiple random seeds.

## B.2   Illustrative example

Heteroskedastic data was generated by sampling $x \sim \mathcal{U}(-1, 1)$, and computing the target $y = \cos(5x) + 0.3 * \epsilon_1 + 1.8\sigma(5x)\epsilon_2$, where $\epsilon_1, \epsilon_2 \sim \mathcal{U}(-0.5, 0.5)$, and $\sigma(\cdot)$ is the sigmoid function.

The base predictor was a 2 hidden layer feedforward network with ReLU activations, with layer width of 64 neurons. We trained the network on 100 randomly sampled datapoints, and held out $N$ new datapoints to use for calibration. In the results in the paper, for all methods we targeted $\alpha = 0.1$ with $\delta = 0.05$. We held the base predictor fixed, and optimize the score function

$$s_{\text{optim}}(\boldsymbol{x}, \boldsymbol{y}; \theta) = \frac{|f(\boldsymbol{x}) - \boldsymbol{y}|}{1 + \sigma(\theta_g)u(\boldsymbol{x}, \boldsymbol{y}; \theta_u)} \tag{21}$$

where $\sigma$ is the sigmoid, and $\theta_g$ functions as a gating parameter, and $u(\boldsymbol{x}; \theta)$ is a learned input-dependent uncertainty function defined as

$$u(\boldsymbol{x}, \boldsymbol{y}; \theta_u) = -1 + \text{softplus}(\text{FF}(\boldsymbol{x}, \boldsymbol{y}, \theta_u) + 0.6), \tag{22}$$

where FF is a two-hidden layer feedforward network with $\tanh$ activations and a layer width of 128, and the softplus together with the offsets smoothly maps the output of the neural network to (approximately) $\mathbb{R}_+$.

For the learned model, we optimize the efficiency loss for 2000 steps with a learning rate of 1e-3 and a batch size of 100 samples. For the PAC-Bayes approach, we optimize using an augmented Lagrangian method, using 2000 steps of gradient descent with a learning rate of 1e-3 to solve the unconstrained penalized problem, running 7 outer iterations in which the penalty terms are updated and the unconstrained problem is re-solved. Each round of inner optimization requires around 1 minute of compute. In the results shown in the paper, we optimized only the prior mean $\boldsymbol{\mu}$, and held the prior variance fixed at $\boldsymbol{\sigma}^2 = 0.02/\sqrt{F}$ where $F$ is the `fan_in` for that parameter.

## B.3   Corrupted MNIST

For the corrupted MNIST experiments, we use 7000 images from the MNIST train split to train a base predictor with the LeNet-5 architecture. For the calibration and test data, we use the held-out test split from MNIST and apply random rotation, uniform between $-30°$ and $30°$ and Gaussian noise with standard deviation 1.3. We save these transformed data, and then split this data data into two disjoint subsets for the calibration and test sets respectively, using a different split for each choice of random seed. We evaluate each approach on 1000 test samples.

Here, the score function for all methods is fixed to be the negative log softmax output by the base predictor, but for the learned and PAC-Bayes methods, we consider optimizing the fully-connected portion of the network. The efficiency loss we use is a differentiable approximation of the output set

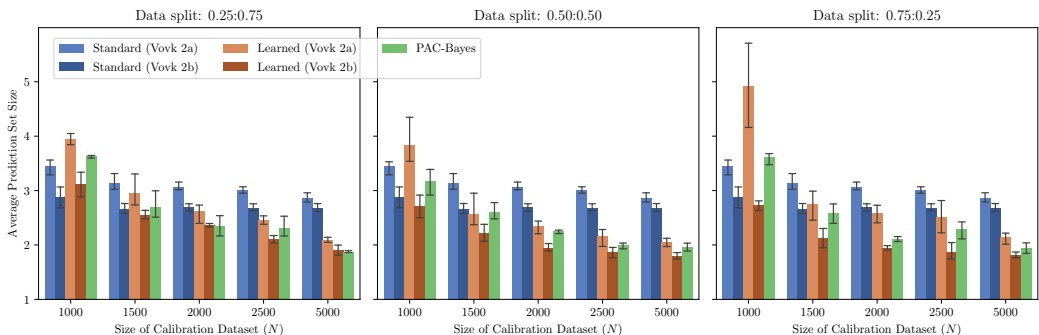

Figure 5: Average prediction set size on held out test data as a function of calibration set size $N$ for different data split ratios.

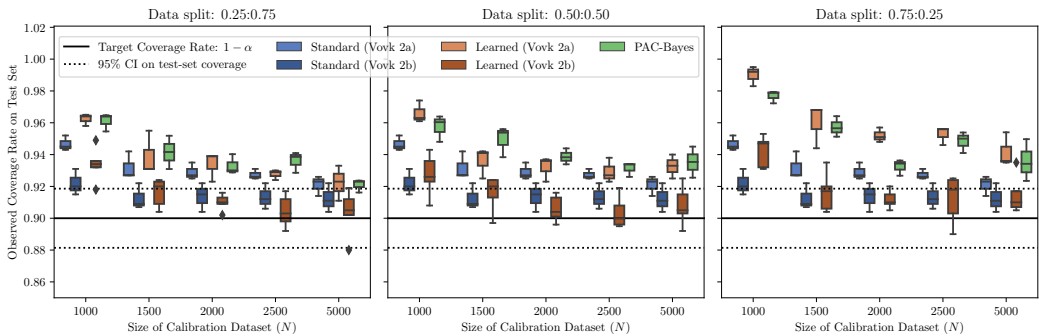

Figure 6: Average coverage rate on held-out test data as a function of calibration set size $N$ for different data split ratios. Box plot shows variation in results over different seeds.

size. Note while the actual set-size of the resulting prediction sets is not Lipschitz continuous w.r.t. the threshold $\tau$, the smoothed loss is, as it is the sum of $K$ sigmoid functions which are themselves Lipschitz continuous, where $K$ is the number of classes. Here, we optimize the efficiency using the same optimization parameters as in the illustrative example. For the PAC-Bayes method, we initialize the prior with mean equal to the base predictor's original trained weights, and a variance with scale $\boldsymbol{\sigma}^2 = 0.01/\sqrt{F}$ where once again, $F$ corresponds to the is the `fan_in` for the layer that the parameter is in. In the body, we report results obtained by optimizing both the prior mean and variance to minimize the efficiency objective on $D_0$ before optimizing the posterior subject to the KL constraint on $D_N$. We found that this led to better results, as selecting the variance using the simple initialization strategy we used in the illustrative example includes too many $\theta$ values which lead to inefficient sets, hindering the quality of the posterior that can be found within the KL constraint.

### B.3.1 Additional Results

In Figure 5, we show the complete set of prediction set size vs calibration dataset size results, for all choices of dataset split. In general, the PAC-Bayes approach yields equal or better sets than baselines across the choice of data split. Figure 6 shows the achieved test-time coverage rates for all methods. All methods target a generalization coverage rate of at least $1 - \alpha = 0.9$, but due to a finite test set size $N_{\text{test}} = 1000$, the empirical coverage rate may appear lower. We show 95% confidence intervals for the empirical mean of 1000 draws from a Bernoulli random variable with probability 0.9. For our results to be consistent with the desired PAC guarantee, failures would only happen in the case of (a) our PAC guarantee failing to hold and the true coverage rate being lower than $1 - \alpha$, which should happen with probability less than $\delta$, or (b) the coverage rate being higher than $\alpha$ but the empirical mean on the test set being lower than the 95% CI for $\alpha = 0.9$, which should occur with probability

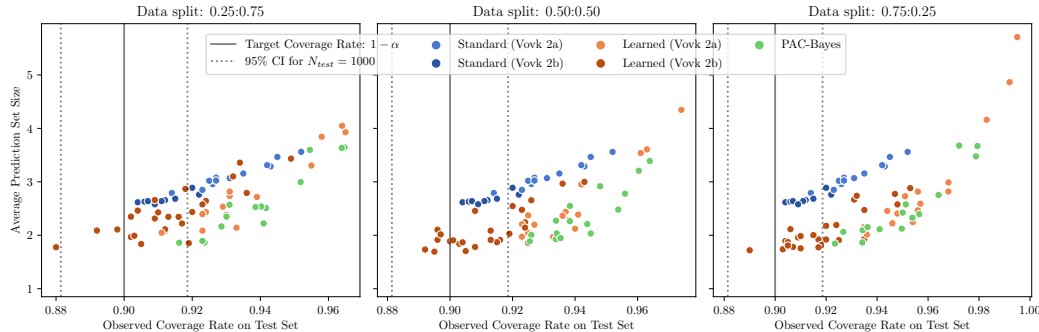

Figure 7: Coverage vs efficiency plots on the MNIST experiment. There is an inherent trade-off between coverage and efficiency (larger sets inherently are more likely to contain the ground truth). Our PAC-Bayes approach consistently lies on the Pareto Frontier in these plots (high coverage and small set size).

less than $0.05$. Indeed we see that all but one trial for the Learned (Vovk 2b) baseline yields an empirical coverage less than the 95% CI. We performed 30 total trials per method, so this observation is consistent with our PAC guarantee and finite test-set size. We find that the PAC-Bayes method is not significantly over-conservative compared to baselines using Vovk 2a to select the empirical coverage targets. However, Vovk 2b leads to much tighter results, indicating room for improvement in our PAC-Bayes results. The difference between the optimized methods and the standard ICP coverage rate is more pronounced for the 0.75:0.25 data split because the optimized method are using only a quarter of the data to guarantee test time coverage, while the standard method uses the whole dataset and thus can calibrate to a lower empirical miscoverage rate $\hat{\alpha}$ following (4).

We also visualize efficiency (prediction set size) as a function of coverage in Figure 7. Indeed, these two quantities are linked: achieving higher coverage for a fixed score function requires increasing set size, coming at the cost of efficiency. The PAC-Bayes results lie on the Pareto frontier on this trade-off, highlighting that our method is able to learn good score functions that yield efficient test-sets on the test set. Future work tightening the generalization bound has the potential to improve practical utility by reducing over-conservatism and allowing calibrating to lower empirical coverage levels.

In Figure 8, we show the impact of different choices of prior optimization strategy, comparing no optimization (using the prior as initialized, using the whole calibration dataset to perform constrained optimization of the posterior) against both mean-only ($\boldsymbol{\mu}$), and mean-var ($\boldsymbol{\mu}, \boldsymbol{\sigma}^2$). As can be seen, a crude, naive strategy of arbitrarily defining a prior by choosing a fixed variance around the initialization of the weights does not work well for this problem setting, highlighting the value of using some data to tune the prior. Furthermore, we find that optimizing both $\boldsymbol{\mu}$ and $\boldsymbol{\sigma}^2$ using a fraction of the data outperforms optimizing only $\boldsymbol{\mu}$, for all choices of data split.

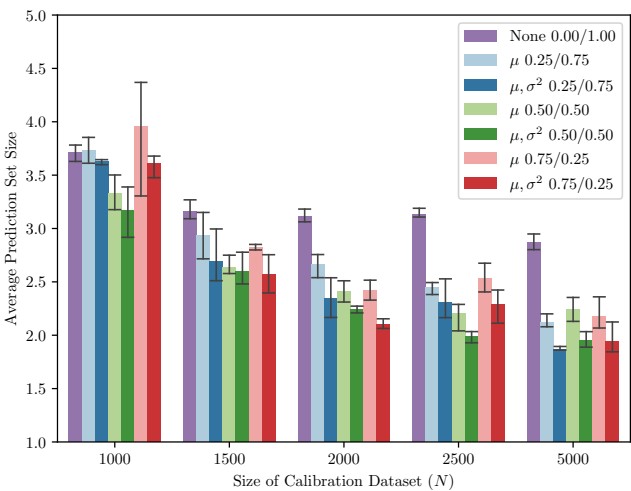

Figure 8: Average test-time prediction set size as a function calibration set size $N$ for different prior optimization strategies and data split between $D_0$ and $D_N$.

