# OpenReview forum: "PAC-Bayes Generalization Certificates for Learned Inductive Conformal Prediction"
_NeurIPS.cc/2023/Conference — NeurIPS 2023 poster_

### Official Review · Reviewer_XWYk · 2023-06-28

**Soundness:** 2 fair
**Presentation:** 3 good
**Contribution:** 2 fair
**Rating:** 3
**Confidence:** 4

**Summary:**

Inductive Conformal Prediction (ICP) provides a coverage guarantee of constructed prediction sets, while does not provide any guarantee on the efficiency of prediction sets. The efficiency depends on a conformity score function, and direct approaches optimize the score function to minimize the efficiency of prediction sets in two steps: (1) given a hold-out set from a calibration set, optimize the score function, and (2) given another hold-out set from a calibration set, run an ICP algorithm. However, the direct approaches do not provide a generalization guarantee for efficiency. The proposed approach combines ICP and the PAC-Bayes to provide generalization guarantees for both coverage and efficiency. In particular, the paper provides novel generalization bounds for coverage and efficiency and proposes an algorithm from them, where the efficacy is evaluated on regression and image classification problems.

**Strengths:**

**Originality**: The paper proposes new generalization bounds on coverage and efficiency.

**Quality**: The proposed algorithm is supported by theorems, making the algorithm rigorous.

**Clarity**: I like the paper structure. It first provides the main theories, and then explains an algorithm with practical details (appreciate it).

**Significance**: I believe the PAC-Bayes interpretation introduces an interesting, novel view on conformal prediction, which could trigger interesting CP algorithms.


**Weaknesses:**

Combining PAC-Bayes and ICP is very interesting, though I have a few concerns.

1. It is not easy to find the benefit of the bound on efficiency (i.e., Thm2). This is novel, but in Algorithm 1, it also uses the first term of (8). What is the algorithmic benefit of the efficiency bound Thm2?
2. I assume that this paper uses the PAC-style ICP (4) as the baselines (i.e., “the standard ICP approach” and “a  learned ICP approach”. However, (4) is known to be loose (as clearly mentioned in Vovk [2012]. In the same Vovk’s paper, it also has a tighter ICP (i.e., Proposition 2b in Vovk [2012]) and its tightness (and efficiency) is also well-demonstrated in deep learning applications in [R1]. If the authors use the tiger PAC-style ICP, would “the standard ICP” and “a learned ICP” baselines be possible to be better than the proposed approach in Figure 4?
3. It is unclear what the benefit of PAC-Bayes is in achieving an efficient prediction set. Based on Line 50 (i.e., “our framework allows us to utilize the entire calibration dataset…”), I initially thought that the proposed approach fully utilizes the calibration set (while direct approaches require splitting calibration sets into two). However, the proposed approach also needs to split the calibration sets for prior learning (i.e., Line 238). At this point, I was unsure of the benefit of combining PAC-Bayes with ICP. My guess is simultaneous optimization of a score function distribution Q and threshold \hat{tau} is one factor for achieving a better efficiency than the baselines, but I want to hear the author's clarification.
4. The nonconformity score function of the standard ICP in regression is quite weak, which misleads readers. The score function with only the residual (i.e., Line 282) is not recommended as it does not encode per sample uncertainty (as also mentioned in the paper). Instead, a variance normalized score function (i.e., (8) in [R2], 3.6 in [R1], or 2.2.1 in [R3]) is a better choice, where the variance is simply trained via a training set. Given this better score function, I don’t think “the standard ICP” result in Figure 3 is as poor as shown in the paper.


[R1] https://arxiv.org/abs/2001.00106

[R2] http://proceedings.mlr.press/v91/vovk18a/vovk18a.pdf

[R3] https://arxiv.org/abs/1612.01474


**Questions:**

Each question is associated with each weakness in Weaknesses.

1. What is the algorithmic benefit of the efficiency bound in Thm2?
2. If the authors use the tiger PAC-style ICP (i.e., Proposition 2b in Vovk [2012]), would “the standard ICP” and “a learned ICP” baselines be possible to be better than the proposed approach in Figure 4?
3. Could you clarify on the benefit of PAC-Bayes in achieving an efficient prediction set?
4. If the authors use the variance normalized score function, do we still observe the same limitation of “the standard ICP” in Figure 3?


**Limitations:**

The limitations are clearly stated in Conclusion, and I agree with that.

---

> ### Author Rebuttal · Authors · 2023-08-10
>
> Thank you for your thorough review and valuable feedback. We address each of your questions below:
> - *Efficiency bound (Theorem 2):* As you noted, we don’t directly optimize the generalization bound in Theorem 2 in our algorithm. This is due to the fact that the optimizable part of the second term in (8) is just the KL divergence, and the coverage constraint already regularizes the KL value. We found that in practice, optimizing through the bound did not improve performance, and excluding it led to a simpler algorithm. However, we did use the generalization bound when comparing between hyperparameter settings, reporting test-time results on only the hyperparameters that achieved the best generalization bound according to Theorem 2. We use a union bound argument, using $\delta’=\delta / K$ when evaluating bounds, where K is the number of hyperparameter choices.
> - *Vovk Prop 2b*: First, we note that the mapping from our notation to that in [Vovk 2012] is $\delta \rightarrow \delta, \hat{\alpha} \rightarrow \epsilon$, and $\alpha \rightarrow E$. We will use our notation to discuss comparisons between our choices and those in Vovk 2012.
>
>   - Indeed for the learned and standard ICP baselines in our experiments, we follow the recommendation of [Vovk 2012] and use the result of Prop 2a as a recipe to construct a $(\alpha, \delta)$-valid conformal predictor. Specifically, given $\alpha$ and $\delta$, this recipe involves constructing a set predictor to achieve $1-\hat{\alpha}$ coverage on the calibration dataset, where $\hat{\alpha} = \alpha - \sqrt{(-\ln \delta)/(2n)}$. Then, we apply Prop 2b once using the chosen $\hat{\alpha}$ and $\delta$ to guarantee $(\alpha,\delta)$-validity of the resulting predictor.
>
>   - We implemented the Prop 2b bound and used it similarly to [R1] to find the largest $\hat{\alpha}$ for which the equation in Prop 2b holds for our desired $\alpha$ and $\delta$, and include the results in the attached pdf under the label (Vovk 2b). While the less conservative $\hat{\alpha}$ leads to tighter prediction sets, empirically we find that the coverage guarantee is violated, especially for the learned version. Indeed, applying a binomial test on the rate of test-time coverage violation for the Learned (Vovk 2b) trial evaluated on N=2500 calibration datapoints split at a ratio of 0.5/0.5 over 6 independent seeds, we obtain a p-value of 0.0327 for the null hypothesis that the probability of failure (coverage < $1- \alpha = 0.90$) is less than $\delta=0.05$, indicating we should reject the null hypothesis that the coverage guarantee is valid for this trial. We are investigating the reason for this discrepancy, to identify if there is an aspect of our approach which breaks the assumptions required by Prop 2b. In any case, we thank the reviewer for highlighting this tighter bound, and we will investigate if a similar tightening can be translated to a PAC-Bayes analysis.
>
> - *Benefit of PAC-Bayes:* Prior works learning score functions for conformal prediction use only a portion of the calibration dataset to optimize the score function to achieve good efficiency in prediction sets whose threshold $\tau$ is computed on the same portion of the calibration dataset. However, especially in the low-data regime, this can lead to cases where the score function over-fits to the portion of calibration data – then, when recalibrating the model (i.e. re-computing the threshold $\tau$ on the remaining portion of the dataset), the score function may behave erratically, leading to a different $\tau$ value and potentially worse efficiency. In contrast, the PAC-Bayes approach allows further optimization of the score function on the second part of the dataset. In this way, it can mitigate the impacts of overfitting in the low-data regime.
> - *Variance-normalized score function:* Our aim in this work is to address situations where calibration data faithfully representing test-time conditions is limited, and one may want to simultaneously fine-tune models on this data while also producing calibrated prediction sets. To illustrate this set-up in a simplified illustrative example, we considered a (synthetic) example, where the pretrained model did not produce variance outputs, but the heteroskedasticity in the calibration dataset was information, which, if effectively incorporated, could lead to more efficient prediction sets. Our goal with the illustrative model was not to demonstrate the real-world use of the method, but rather to highlight that our approach can more effectively learn new concepts and produce calibrated prediction sets from a limited set of calibration data. While in practice, training a model to produce variance outputs would yield tighter prediction sets to begin with, this would not illustrate the utility of further fine-tuning a model on calibration data. Indeed, the utility of this approach is highest when calibration data is a scarce resource, e.g., when we only have a limited set of samples from a shifted test data-distribution. We will make this point clearer in the revision.

---

> > ### Comment · Reviewer_XWYk · 2023-08-15
> > **Discussion**
> >
> > Thanks for the detailed responses. I wish to initiate the discussion.
> >
> > - *Efficiency bound (Theorem 2)*: the answer is a bit confusing. Based on the answer, the second term is useful in hyper-parameter selection but not useful in test-time performance, which sounds inconsistent. Usually, we use a loss function in hyper-parameter selection and the best model is also good w.r.t. the same loss function. Can you clarify the counter-intuitive usage? What is the hyper-parameter selection procedure? What is $K$ (which is not appeared in the paper)? Can I say that you're using $\delta'$ in Algorithm 1, instead of $\delta$?
> >
> > - *Vovk Prop 2b*: To my understanding (based on the empirical coverage violation), the author's implementation of *Vovk Prop 2b* has some issues. I'd recommend to draw the whisker plot such that the open interval from the bottom tip of the whisker plot contains $100\delta$ % of samples --- this is the meaning of $\delta$. If the bottom of the whisker plot is still below of the desired coverage rate, then it's the sign of implementation bugs. I think comparison to this tighter baseline should be provided to justify the efficacy of the proposed approach.
> >
> > - *Benefit of PAC-Bayes:*: I'm not quite convinced the provided intuitive explantation. For example, consider the claim of "the score function over-fits to the portion of calibration data – then, when recalibrating the model". For the low-data regime, I agree that the score function overfits, but during the recalibration, the low-sample size is still accounted (i.e., Vovk Prop 2a/2b are a function of the sample size). I think we need to see empirical benefits of PAC-Bayes compared to the **fixed** "Learned (Vovk 2b)" first for further discussion.
> >
> > - "Variance-normalized score function": I guess my concern was not addressed. In other words, my point is that "Learned ICP + Recal" essentially wastes calibration samples and that's why its efficiency is poor (i.e., a weaker baseline). A better way is considering the variance-normalized score function and recalibration with a **full** calibration set (i.e., a stronger baseline). This should be the starting point. Given this stronger baseline, I would be excited if the PAC-Bayes results outperform the stronger baseline.
> >
> > In short, I found the baseline used in this paper is weak in two senses: (1) a weak scoring function (i.e., the paper does not use the variance-normalized score function) and (2) a weak ICP PAC bound (i.e., the paper does not use Vovk Proposition 2b). This does not convince myself whether the PAC-Bayes can serve as an exciting new bound. So, I'll maintain my score at this point, but I hope to see a better PAC-Bayes bound!

---

> > > ### Author Response · Authors · 2023-08-21
> > > **Response to questions**
> > >
> > > Thanks for starting the discussion. Below are our responses to your questions, which we hope provide some clarity.
> > >
> > > *Efficiency bound* We wish to clarify that the model with the lowest efficiency bound was in general had the best test-time performance; hence it served well as a loss to select between hyperparameter settings. Our claim in our comment was addressing why we did not use the full bound in our optimization procedure: we found that differentiating through the full bound during optimization did not significantly improve performance, as the constraint imposed by Theorem 1 ensured the KL divergence remained small.
> > >
> > > $K$ is the number of hyperparameter options considered. For the hyperparameter selection, we considered choosing a value for $\hat{\alpha}$ between $K=5$ values between 0.02 and 0.08. To do so, we optimized a score function for each setting on the same dataset using $\delta’ = \delta/K = 0.05/5 = 0.01$. Then, we evaluated the efficiency bound for each of the 5 models using this value of $\delta’$ in place of $\delta$. In this way, the bound holds for each of the models independently with probability $1 - \delta’$, i.e. $1-\delta/K$. Therefore, by the union bound, the probability of all $K=5$ bounds holding jointly must be greater than $1-\delta$, which is our desired guarantee. Thank you for pointing out that our discussion of hyperparameter selection can be further clarified; we will update it in the revised manuscript.
> > >
> > > *Benefit of PAC-Bayes* While the recalibration procedure accounts for the low sample size by choosing a more conservative choice of $\hat{\alpha}$, this is to ensure that the *coverage* guarantees still hold on test-data. The sample-size considerations in Vovk 2a and 2b make no claims to ensure the efficiency of the sets (e.g. their size) also generalizes well. This is what we are referring to by overfitting – we might learn a score function that achieves efficient set sizes on the particular data points that are in the calibration set, but produce inefficient sets on test data. In our problem setting, our aim to learn sets whose coverage *and* efficiency both generalize well to the test data distribution. We hope this provides some clarity on the novelty of our approach over prior work.
> > >
> > > *Variance-normalized score function* Indeed, the Learned ICP + Recal appears to “waste” samples by only learning the variance normalization on part of the calibration data, and using the rest to select the threshold. However, prior work **requires** this recalibration on a **held-out** dataset in order to provide the PAC guarantee on test-time coverage. If we were to use the full calibration dataset to learn the variance normalization, and then recalibrate (i.e. compute the threshold $\tau$) on the same dataset, the guarantees on test-time coverage would not hold. This is because after optimization, the score function is no longer independent of the calibration dataset. This apparent “waste” is precisely the limitation of prior work we aim to address; by leveraging PAC-Bayes theory, we can use the **full** dataset to optimize the score function for efficiency, without losing test-time guarantees.
> > >
> > > Nevertheless, we agree that a comparison to this version would help make this point more clear, and we will update our manuscript to include these results.
> > >
> > > *Vovk 2b*  Thank you for your suggestion. We will update the visualization for the results we include in the revised version.

---

### Official Review · Reviewer_6Bn9 · 2023-07-04

**Soundness:** 3 good
**Presentation:** 3 good
**Contribution:** 2 fair
**Rating:** 5
**Confidence:** 3

**Summary:**

This work first derives coverage and efficiency generalization bound using PAC-Bayes for inductive conformal prediction. Furthermore, a practical algorithm, basing bayesian learning and conformal training, is proposed to learn efficient nonconformity score functions. In experiment section, the proposed algorithm is shown to outperform two baselines (original ICP and learned ICP) in some scenarios.

**Strengths:**

1, This work is mathematically solid, by combining PAC-Bayes theory with inductive conformal prediction. Particularly, it derives a efficient loss bound using PAC-Bayes for ICP.
2, In the experiment section, the proposed method outperforms other two baselines (original ICP and learned ICP) in many scenarios, demonstrating high efficiency, especially when limited calibration data is available.
3, The manuscript is well-written; and proposed method is mathematically sound.

**Weaknesses:**

1, In the abstract, it is claimed that "the entire calibration dataset" can be utilized. In the practical algorithm, splitting the calibration dataset is still needed (to derive a good prior). As a result, in my opinion, the mentioned claim is at least controversial.
2, Though not mentioned, the proposed algorithm could be expensive in training and inference, as multiple models need to be sampled.
3, The empirical evaluation is insufficient. Only one synthetic dataset and one practical dataset are used. It would be appreciated if experiments on more practical datasets. With more empirical results, it could be more convincing to claim that the proposed method produces more efficient prediction sets.

**Questions:**

1, How are the two derived bounds connected to the proposed algorithm?

**Limitations:**

1, The computation complexity might be so high that the proposed method cannot trivial extend to larger datasets and models.
2, Without good prior, the derived bounds might be too loose.

---

> ### Author Rebuttal · Authors · 2023-08-10
>
> Thank you for your insightful and helpful comments. We address your criticisms and questions below:
> - *Data-splitting:* As you point out, in the experiments we split the data, using part of it to tune the prior, and the rest to simultaneously tune the posterior and also achieve test-time coverage guarantees. However, both splits of data are used to optimize the score function. This is in contrast to prior work in learned conformal prediction, which optimizes the score function on one set of data, and then holds it fixed when using the remainder to compute the threshold needed to achieve test-time coverage guarantees.
> - *Sampling multiple models:* Indeed, PAC-Bayes style generalization bounds require reasoning over stochastic models rather than individual hypotheses. However, at test time, each data point is only evaluated under a single model: test-time computational complexity at a single-sample level is therefore similar to a non-stochastic approach. Indeed, our guarantee on test-time coverage holds in expectation over samples drawn from the posterior and data drawn from the data distribution, so as long as each new input is evaluated using a new sample, our guarantee holds. However, one important limitation is that we require recomputing the threshold $\tau$ for each new sample. For small calibration sets and models, this can be done in parallel at test time. Alternatively, one can pre-sample K models from the posterior, and use the calibration data to pre-compute the threshold $\tau$ for each sample. This offloads the computational burden from test-time evaluation, but adds to the memory burden, as multiple samples of model parameters need to be stored. We will emphasize this limitation in the updated manuscript and more clearly discuss this tradeoff.
> - *Empirical evaluation:* We believe the core of this paper is our theoretical contributions. In particular, in contrast to standard ML settings where the loss is defined per-datapoint, in conformal prediction, the efficiency and coverage on a particular data point depends on the threshold, which itself is computed as a function of the entire calibration dataset. This dependency precludes direct application of standard PAC-Bayes proofs, and requires a novel theoretical approach, which we consider the main contribution of our work. As such, while we agree that more experiments would always be better, we would like to stress that our goal in the empirical evaluation was not to demonstrate the immediate practical applicability of this approach, but rather to highlight the potential of our theoretical contributions. We believe our experiments demonstrate that our approach can already yield modest benefits in terms of efficiency in the low-data regime, and these benefits have the potential to become more pronounced thanks to future advances in the tightness of PAC bounds and advances in prior and posterior selection.
> - *How are generalization bounds used?* We produce two main generalization bounds. The bound on coverage (Theorem 1) is used to derive Corollary 2.1, which is used directly in the algorithm to compute the KL budget allowed for optimizing the posterior as a function of the desired test-time coverage guarantee. The bound on efficiency (Theorem 2) is used to select between different hyperparameter settings: we choose the model with the best bound on generalization efficiency among K hyperparameter choices. In our experiments, we evaluate these bounds using $\delta’ = \delta/K$, ensuring through a union bound argument that the generalization bounds hold for the best-performing model with probability greater than $1-\delta$.

---

> > ### Comment · Reviewer_6Bn9 · 2023-08-16
> >
> > Thanks for addressing my questions and concerns!
> >
> > I agree that the core of this paper is the theoretical contribution, which is novel by itself.
> >
> > However, since the results are only on simple datasets, it is still unclear to me how useful these bounds would be, considering that this method would induce higher computational complexity.
> >
> > As a result, I will raise my score to borderline accept.

---

### Official Review · Reviewer_Akhh · 2023-07-06

**Soundness:** 3 good
**Presentation:** 3 good
**Contribution:** 3 good
**Rating:** 6
**Confidence:** 3

**Summary:**

The efficiency of set-valued predictor is crucial and less explored, this paper use the framework of PAC-Bayes to obtain the generalization bounds on both the coverage and the efficiency of set-valued predictors. The authors also propose a novel algorithm to optimize the efficiency without a separate hold-out dataset. They provide a theoretical guarantee for the proposed method and empirically verify the superiority of the proposed method.

**Strengths:**

- This paper uses the framework of PAC-Bayes to analyze the efficiency of inductive conformal prediction, which is novel and less explored.
- The algorithm proposed by this paper does not require a separate hold-out dataset, while still having a theoretical guarantee of coverage.
- This paper is well organized and the presentations are clear to me.

**Weaknesses:**

- To achieve the test-time coverage, the key constraint is shown in Corollary 2.1. However, the authors do not show whether it is satisfied in the experiments.
- In the experiments, the authors show the average size of the prediction set, but I think the marginal coverage evaluated on the test data in the experiments is also important, and does a smaller average size of the prediction set empirically lead to worse coverage?
- In the experiments, the description of the ***learned*** baseline is unclear. Do the authors train the model using the method proposed in [1]? Meanwhile, the authors miss a recent work [2] that aims to output a smaller conformal prediction set with higher conditional coverage. [1] and [2] should be chosen as baselines in the experiments.

[1] David Stutz, Krishnamurthy Dvijotham, Ali Taylan Cemgil, Arnaud Doucet. Learning Optimal Conformal Classifiers. ICLR 2022.

[2] Bat-Sheva Einbinder, Yaniv Romano, Matteo Sesia, Yanfei Zhou. Training Uncertainty-Aware Classifiers with Conformalized Deep Learning. NeurIPS 2022.

**Questions:**

Please see the weaknesses.

**Limitations:**

A limitation of this paper is that the theoretical results and proposed method can only be applied for a stochastic model.

---

> ### Author Rebuttal · Authors · 2023-08-10
>
> Thanks for your detailed feedback. We address your questions below:
> - *Constraint Satisfaction:* The constrained optimization solver we use in our experiment ensures that solutions satisfy Corollary 2.1 by construction. In particular, we solve the constrained optimization problem with an augmented Lagrangian algorithm (approximating the inner optimization with stochastic gradient descent), and only return a solution if it is feasible (i.e., if the solution satisfies Corollary 2.1). We will clarify this in the updated manuscript, and include results on constraint values in the appendix.
> - *Coverage results:* We focused on presenting efficiency results because all approaches yielded similar coverage values that exceeded the desired value; however, we did include marginal coverage results in the supplementary material – we will update the main paper to highlight that these results are included in the appendix. We agree that the relationship between test-time coverage and test-time efficiency is interesting, and we will include these results (shown in the attached pdf, Fig. 2) in the updated version of the manuscript.
> - *Learned baseline:* Indeed, the learned baseline is a reimplementation of ConfTr [Stutz et al, 2022], extended to the regression case by using the radius of the prediction set as an efficiency loss. We will make this more clear in the updated version of the manuscript. We also thank the reviewer for pointing out the work of [Einbinder et al, 2022] – we will update our related work section to discuss it. Indeed, they tackle a similar problem as [Stutz et al, 2022] of training ML models to yield efficient sets after conformalization; however, their approach is limited to a specific nonconformity function designed for classification. In contrast, our work (as well as [Stutz et al, 2022]) works with any nonconformity function and extends beyond classification problems. For this reason, we cannot directly compare against [Einbinder et al, 2022] in our experiments.

---

> > ### Comment · Reviewer_Akhh · 2023-08-14
> >
> > Thanks for the response, and I decide to raise my score to a weak accept.

---

### Official Review · Reviewer_kKqP · 2023-07-07

**Soundness:** 3 good
**Presentation:** 3 good
**Contribution:** 3 good
**Rating:** 6
**Confidence:** 3

**Summary:**

Accurate uncertainty estimation is crucial in building robust and trustworthy machine learning systems. This paper utilizes PAC-Bayes theory and inductive conformal prediction (ICP) to develop a practical algorithm that fine-tunes parameters and score functions using calibration data, while ensuring inference coverage and efficiency. Empirical validation of this work shows that it is effective and outperforms prior methods.

**Strengths:**

The paper presents a well-executed translation from theory to practical algorithm and provides a comprehensive analysis of related work. It also discusses the practical considerations in their proposed algorithm and effectively demonstrates its ability to make efficient predictions and achieve good generalization certificates in two experiments.

**Weaknesses:**

Here are some suggestions for improving the original content:

1. It would be beneficial to provide more information and explanation in section 5.2 to help readers better understand the algorithm.
2. Consider including an experiment that explores the tradeoff between desired coverage and prediction performance. This would provide valuable insights into the algorithm's effectiveness.

**Questions:**

I am interested in understanding the extent to which distribution shift can affect the performance of calibration and its coverage. Adding more noise to MNIST figures can increase the level of uncertainty, making it more challenging to ensure accuracy.

**Limitations:**

The paper identifies various limitations and proposes practical solutions to address them. The significance of this work lies in its contribution to achieving a more reliable and secure ML system. Particularly, it demonstrates its effectiveness in providing assurance to enhance the calibration of uncertainty.

---

> ### Author Rebuttal · Authors · 2023-08-10
>
> Thanks for your review and insightful criticisms and comments. We address specific comments below:
> - *Section 5.2:* We acknowledge that section 5.2 is a very succinct summary of the optimization approach. We will update our manuscript to go into more details of the algorithm, and add a section in the appendix which goes into more detail and reintroduces the concepts from [Stutz et al., 2022] that we re-use in our implementation to make the manuscript more self-contained.
> - *Coverage/Efficiency Tradeoff:* The coverage / efficiency tradeoff is a fundamental tradeoff for all conformal prediction approaches, and we agree that plots showing this tradeoff would be a welcome addition to the results presented in our paper. We plotted this trade-off across all dataset sizes in the attached pdf to the main response (Fig 3.), and we will include it in our updated version. Notably, while all methods see a tradeoff between coverage and efficiency, our PAC-bayes method consistently lies on the Pareto frontier.
> - *Distribution Shifts:* Indeed, distribution shifts can have an impact on calibration. Prior work evaluating uncertainty quantification under distribution shift has demonstrated that neural network uncertainty estimates can struggle to generalize as the data distribution is shifted [1]. Indeed, for this reason, we chose a distribution shift example to study our approach: While a network trained on clean MNIST does produce heuristic uncertainty estimates (through its predicted class probabilities), these estimates are unlikely to generalize well to corrupted data. Hence, if we have a limited amount of data from the test (shifted) domain, we would like to ideally use this data to both tune the network to produce better uncertainty signals as well as to produce calibrated uncertainty sets, as directly using the untuned network could yield loose prediction sets.
> Now, a separate challenge is handling distribution shift *after* the calibration procedure. Robustifying standard conformal prediction to future distribution shift is an active area of research [2], as is literature extending PAC-Bayes to settings where the test-data distribution differs from the training data distribution [3]. We considered this challenge to be out-of-scope for this work, but we wholeheartedly agree that this is an important problem and a worthwhile direction for future extensions of this work.
>
>  [1] Ovadia, Yaniv, et al. "Can you trust your model's uncertainty? evaluating predictive uncertainty under dataset shift." Advances in neural information processing systems 32 (2019).
>
>  [2] Tibshirani, Ryan J., et al. "Conformal prediction under covariate shift." Advances in neural information processing systems 32 (2019).
>
>  [3] Germain, Pascal, et al. "A new PAC-Bayesian perspective on domain adaptation." International conference on machine learning. PMLR, 2016.

---

> > ### Comment · Reviewer_kKqP · 2023-08-15
> > **Thanks for your response.**
> >
> > Dear Authors,
> >
> > Thanks for your response. I have read your figure 3 about Coverage/Efficiency Tradeoff and your clarifications about distribution shifts. I keep my score and support this paper to be accepted.

---

### Official Review · Reviewer_jcN9 · 2023-07-26

**Soundness:** 3 good
**Presentation:** 3 good
**Contribution:** 3 good
**Rating:** 7
**Confidence:** 4

**Summary:**

This paper considers the setting of inductive conformal prediction (ICP). In this setting, given a learned point-predictor, a calibration set and a scoring function is used to obtain a set-valued predictor that contains the correct label with high probability. A drawback of previous approaches is that, for ICP guarantees to be valid, the scoring function has to be fixed, or part of the calibration set needs to be used to learn the scoring function, while the remaining part is used to learn the coverage sets. In order to solve this, the paper proposes to use PAC-Bayesian bounds for self-certified learning, so that the scoring function can be learned from the calibration data, while using the same calibration data to obtain test-time bounds by using generalization guarantees. The approach relies on new bounds, which are similar to known PAC-Bayesian bounds but specialized to the ICP setting, and data-dependent Gaussian priors according to a standard procedure. Numerical results demonstrate the potential of the proposed approach to improve ICP in low-data settings.


**Post-Rebuttal edit**: I read the author rebuttal, which addressed my questions, and consequently updated the rating (as stated below).

**Strengths:**

— The paper proposes a creative and relevant use of the PAC-Bayesian framework for a practical use-case

— The presentation is pedagogic, providing an essentially self-contained summary of the necessary background from ICP and PAC-Bayes

— In addition to just presenting bounds, the paper suggests a well-motivated practical algorithm that is shown to have potential to improve results for certain settings. The interpretation of the PAC-Bayesian bound as providing a "KL budget" is neat.

**Weaknesses:**

— The discussion of the PAC-Bayesian bounds does not make it entirely clear what the relation to prior work is (see questions below).

— The improvements that are seen in the numerical experiments appear to be marginal, and mostly within the error bars. The non-monotonicity in the order between "Learned" and "PAC-Bayes" in Figure 4 (middle) appears to highlight that they are essentially indistinguishable. Also, it was not entirely clear to me how Figure 3 demonstrated an improvement for PAC-Bayes.

**Questions:**

— Theorem 1, in terms of the binary KL, is very similar to the Maurer-Langford-Seeger bound (see, e.g., Maurer (2004)). In Maurer's result, a smaller $B(N)$ is found. However, I don't think that the two arguments of your binary KL are related by an expectation as in Maurer's derivation, which necessitates the path that you take. Is this interpretation correct? I think the paper would benefit by clarifying the relation to these classical PAC-Bayesian bounds.

— The derivation of Theorem 3, once the step has been taken from $\mathcal L$ to $R$, appears to be the same as a standard PAC-Bayesian bound via "sub-Gaussian"-type concentration for bounded random variables. Is there any other difference in terms of the proof?

— In the introduction, you refer to PAC-Bayesian priors as data-independent, which may cause some confusion since you use data-dependent priors in your experiments. Perhaps a note should be made that they can be data-dependent in certain circumstances.

---

Maurer, A Note on the PAC Bayesian Theorem, 2004

___
---

Minor comments:

Eq (5): Should this be averaged over the posterior? This is typically the case for PAC-Bayes bounds.

Line 196: “for all for all”

Line 197: bounuded -> bounded in footnote

Line 326: “we found that optimizing the yielded a poor prior” incomplete sentence

Donkser -> Donsker

Overfull margins in appendix

**Limitations:**

Limitations are adequately discussed. Perhaps it can be made clearer that the proposed approach does not yield particularly significant improvements over the "Learned" baseline.

---

> ### Author Rebuttal · Authors · 2023-08-10
>
> Thank you for your very thorough review and helpful comments! Below are answers to your specific questions:
>
> - *Theorem 1:* This is a great observation. Indeed, in the Maurer-Langford-Seeger bound, the two arguments to the binary KL are the empirical risk and the generalization risk: for a fixed hypothesis, the generalization risk has no dependence on the empirical risk. However, in conformal prediction, the test-time prediction sets depend on both the hypothesis (in this case, the score function) as well as the threshold, which is a function of the sampled data. Indeed, unlike typical loss functions that are defined per-datapoint, coverage and efficiency of conformal predictors depend on the threshold computed as a quantile of the entire calibration dataset. This is the key difference that precludes the direct application of a Maurer style PAC-Bayesian bound, and necessitates a different approach which yields a different B(n) term. We will more explicitly highlight these similarities and differences in our updated manuscript.
> - *Theorem 3:* Yes, our proof follows the standard steps after translating from $\mathcal{L}$ to $R$. We will make this clear in the updated manuscript.
> - *Wording in introduction:* We agree that our original wording is a source of confusion. We will update our writing to clarify that the restriction is only that the prior is chosen independently from the data used to compute the generalization bound.
>
> *Limitations:* We will update our discussion of limitations to make clear that our approach yields modest improvements relative to the learned baseline; we will instead highlight the theoretical contributions as noted above, and the potential of the approach to become more useful with future complementary advances in PAC-Bayes literature. Finally, thank you for pointing out the small errors; we apologize for not catching these in our submission and will correct them in our updated draft.

---

> > ### Comment · Reviewer_jcN9 · 2023-08-14
> >
> > Thank you for your response. All my questions are clarified. As I am positive about this work and potential future extensions, I have increased my rating.

---

### Author Rebuttal · Authors · 2023-08-10

We would like to thank the reviewers for their time and valuable feedback on our manuscript. We are glad the reviewers all found our paper clear, sound, and well-structured. Furthermore, we are glad to hear that the reviewers appreciated the novelty of our approach in applying PAC-Bayes generalization bounds to the setting of conformal prediction.

First, we would like to emphasize that our primary goal, as reviewer jcN9 highlighted, is to make a theoretical contribution which allows **self-certified learning of conformal predictors**. Indeed, all prior work on optimizing conformal predictors using data required heuristically splitting data and using part of the data to tune the model and score function parameters, and the rest to compute the threshold needed to achieve the desired test-time coverage guarantees. Our work uses PAC-Bayes generalization theory to show that data-splitting is **not** strictly required to achieve test-time coverage: instead, a coverage guarantee can be achieved by constraining the data-driven optimization in terms of KL divergence to a prior. We would like to reiterate that these results are novel, and more than a direct application of standard PAC-Bayes results: unlike typical loss functions which are defined per-datapoint, the coverage and efficiency of the prediction for a single input depend on the entire calibration dataset, since the threshold defining the prediction sets is computed as an empirical quantile. Handling this required novel extensions to classical PAC-Bayes theory.

Next, we would like to address shared concerns regarding the empirical evaluation of our practical algorithm. Many reviewers pointed out that in our evaluation, we considered using a portion of the available data to first optimize a prior, before then performing a second phase of optimization wherein we retain generalization guarantees on coverage and efficiency by enforcing a KL constraint to the (learned) prior. Indeed, as we state in the paper, this is a common practice for many works applying PAC-Bayes techniques to neural network models: while generalization bounds on loss are valid for any fixed prior, state-of-the-art methods achieving generalization performance competitive with empirical risk minimization (ERM) often use a portion of data to tune a prior [1,2]. Admittedly, this has the consequence of reducing the data that is used to compute the threshold for the conformal prediction sets. Nevertheless, we would like to stress that, in contrast to the data-splitting employed in the learned ICP baseline (and prior works), our approach uses **all** available data to optimize the score function and model parameters, while existing approaches only optimize parameters on one fraction of the dataset. Specifically, we use the first part to optimize the prior for the score function, and then use the second part to optimize the posterior. Earlier data-splitting techniques hold the score function fixed in the second step. Finally, we believe that future advances in applying PAC-Bayes to neural networks, both in terms of prior selection as well as posterior representation, will be complementary to the techniques in our paper and help enhance the practical utility of the algorithm.

Overall, we believe our core theoretical contributions are novel, and of interest to the broader NeurIPS community. Furthermore, we believe our experiments demonstrate that even with a very restrictive class of prior and posterior (diagonal Gaussian), our theoretical contributions yield an algorithm for learning conformal predictors which yields efficient prediction sets in practice and guaranteed test-time coverage, demonstrating the potential of the approach to be useful in the low-data regime. We refer the reviewers to our individual responses for response to other questions raised in the reviews.

[1] Ambroladze, Amiran, Emilio Parrado-Hernández, and John Shawe-Taylor. "Tighter pac-bayes bounds." Advances in neural information processing systems 19 (2006).
[2] Perez-Ortiz, Maria, et al. "Learning PAC-Bayes priors for probabilistic neural networks." arXiv preprint arXiv:2109.10304 (2021).

---

### Decision · Program_Chairs · 2023-09-21

**Decision:**

Accept (poster)

**Comment:**

This meta review is based on the reviews, the authors rebuttal and the discussions with the reviewers, discussions with the SAC, and ultimately my own judgement on the paper. There was a consensus that the paper contributes sound and interesting contributions to the PAC-Bayes theory applied to conformal predictors. I feel this work deserves to be featured at NeurIPS and will attract interest from the community. I would like to personally invite the authors to carefully revise their manuscript to take into account the remarks and suggestions made by reviewers. Congratulations!